# OFF-POLICY EVALUATION WITH DEEPLY-ABSTRACTED STATES

## ABSTRACT

Off-policy evaluation (OPE) is crucial for assessing a target policy's impact offline before its deployment. However, achieving accurate OPE in large state spaces remains challenging. This paper studies state abstractions – originally designed for policy learning – in the context of OPE. Our contributions are three-fold: (i) We define a set of irrelevance conditions central to learning state abstractions for OPE, and derive a backward-model-irrelevance condition for achieving irrelevance in (marginalized) importance sampling ratios by constructing a time-reversed Markov decision process (MDP). (ii) We propose a novel iterative procedure that sequentially projects the original state space into a smaller space, resulting in a deeply-abstracted state, which substantially simplifies the sample complexity of OPE arising from high cardinality. (iii) We prove the Fisher consistencies of various OPE estimators when applied to our proposed abstract state spaces.

## 1 INTRODUCTION

**Motivation.** Off-policy evaluation (OPE) serves as a crucial tool for assessing the impact of a newly developed policy using a pre-collected historical data before its deployment in high-stake applications, such as healthcare (Murphy et al., 2001), recommendation systems (Chapelle & Li, 2011), education (Mandel et al., 2014), dialog systems (Jiang et al., 2021) and robotics (Levine et al., 2020). A fundamental challenge in OPE is its "off-policy" nature, wherein the target policy to be evaluated differs from the behavior policy that generates the offline data. This distributional shift is particularly pronounced in environments with large state spaces of high cardinality. Theoretically, the error bounds for estimating the target policy's Q-function and value decrease rapidly as the state space dimension increases (Hao et al., 2021a; Chen & Qi, 2022). Empirically, large state space significantly challenges the performance of state-of-the-art OPE algorithms (Fu et al., 2020; Voloshin et al., 2021).

Although different policies induce different trajectories in the large ground state space, they can produce similar paths when restricted to relevant, lower-dimensional state spaces (Pavse & Hanna, 2023). Consequently, applying OPE to these abstract spaces can significantly mitigate the distributional shift between target and behavior policies, enhancing the accuracy in predicting the target policy's value. This makes state abstraction, designed to reduce state space cardinality, particularly appealing for OPE. However, despite the extensive literature on studying state abstractions for policy learning (see Section 2 for details), it has been hardly explored in the context of OPE.

**Contributions.** This paper aims to systematically investigate state abstractions for OPE to address the aforementioned gap. Our main contributions include:

1. Introduction of a set of irrelevance conditions for OPE, and derivation of a backward-model-irrelevance condition for state abstractions to achieve irrelevance in marginalized importance sampling ratios by constructing a time-reversed Markov decision process (MDP, Puterman, 2014) that swaps the future and past.
2. Development of a novel iterative procedure to sequentially compress the state space. Specifically, within each iteration, our algorithm consistently produces a state space that is either smaller in size or remains the same. Through its iterative nature, the proposed approach produces a deeply-abstracted state space, which substantially reduces the sample complexity of OPE.
3. Validations of various OPE methods when applied to the proposed abstract state spaces.

**Organization**. The rest of the paper is structured as follows. Section 2 is dedicated to the literature review of related works. MDP-related notions and OPE methodologies relevant to our proposal are

recalled in Section 3. Our proposed state abstractions for OPE are presented in Section 4. Section 5 conducts numerical experiments to demonstrate the efficiency of our approach.

## 2 RELATED WORK

Our proposal is closely related to OPE and state abstraction. Additional related work on confounder selection in causal inference is relegated to Appendix A.

**Off-policy evaluation**. OPE aims to estimate the expected return of a given target policy, utilizing historical data generated by a possibly different behavior policy (Dudík et al., 2014; Uehara et al., 2022). The majority of methods in the literature can be classified into the following three categories:

1. **Value-based methods** that estimate the target policy's return by learning either a value function (Sutton et al., 2008; Luckett et al., 2019; Li et al., 2024) or a Q-function (Le et al., 2019; Feng et al., 2020; Hao et al., 2021b; Liao et al., 2021; Chen & Qi, 2022; Shi et al., 2022) from the data.
2. **Importance sampling (IS) methods** that adjust the observed rewards using the IS ratio, i.e., the ratio of the target policy over the behavior policy, to address their distributional shift. There are two major types: sequential IS (SIS, Precup, 2000; Thomas et al., 2015; Hanna et al., 2019; Hu & Wager, 2023) which employs a cumulative IS ratio, and marginalized IS (MIS, Liu et al., 2018; Nachum et al., 2019; Xie et al., 2019; Dai et al., 2020; Yin & Wang, 2020; Wang et al., 2023) which uses the MIS ratio to mitigate the high variance of the SIS estimator.
3. **Doubly robust methods** or their variants that employ both the IS ratio and the value/reward function to enhance the robustness of OPE (Zhang et al., 2013; Jiang & Li, 2016; Thomas & Brunskill, 2016; Farajtabar et al., 2018; Kallus & Uehara, 2020; Tang et al., 2020; Uehara et al., 2020; Shi et al., 2021; Kallus & Uehara, 2022; Liao et al., 2022; Xie et al., 2023).

However, none of the aforementioned works studied state abstraction, which is our primary focus.

**State abstraction.** State abstraction aims to obtain a parsimonious state representation to simplify the sample complexity of reinforcement learning (RL), while ensuring that the optimal policy restricted to the abstract state space attains comparable values as in the original, ground state space. There is an extensive literature on the theoretical and methodological development of state abstraction, particularly bisimulation — a type of abstractions that preserve the Markov property in the abstracted state (Singh et al., 1994; Dean & Givan, 1997; Givan et al., 2003; Ferns et al., 2004; Ravindran, 2004; Jong & Stone, 2005; Li et al., 2006; Ferns et al., 2011; Abel et al., 2016; Wang et al., 2017; Castro, 2020; Allen et al., 2021; Abel, 2022). In particular, Li et al. (2006) analyzed five irrelevance conditions for optimal policy learning. Unlike the aforementioned works that focus on policy learning, we introduce irrelevance conditions for OPE, and propose abstractions that satisfy these irrelevant properties. Meanwhile, the proposed abstraction for achieving irrelevance for the MIS ratio resembles the Markov state abstraction developed by Allen et al. (2021) in the context of policy learning, while relaxing their requirement for the behavior policy to be Markovian.

More recently, Pavse & Hanna (2023) made a pioneering attempt to study state abstraction for OPE, proving its benefits in enhancing OPE accuracy. However, they primarily focused on MIS estimators. In contrast, our theoretical analysis applies to a broader range of OPE estimators, covering all three aforementioned categories. Moreover, their abstraction did not achieve MIS-ratio irrelevance. Nor did they implement the iterative procedure.

Lastly, state abstraction is also related to variable selection (Kolter & Ng, 2009; Geist & Scherrer, 2011; Geist et al., 2012; Nguyen et al., 2013; Fan et al., 2016; Guo & Brunskill, 2017; Shi et al., 2018; Zhang & Zhang, 2018; Qi et al., 2020; Hao et al., 2021a; Ma et al., 2023) as well as representation learning for both policy learning (see e.g., Gelada et al., 2019; Zhang et al., 2020; Uehara et al., 2021b) and OPE (see e.g., Wang et al., 2021; Chang et al., 2022; Ni et al., 2023; Pavse & Hanna, 2024).

## 3 PRELIMINARIES

In this section, we first introduce some key concepts relevant to OPE in RL, such as MDP, target and behavior policies, value functions, IS ratios (Section 3.1). We next review state abstractions for optimal policy learning (Section 3.2), alongside with four prominent OPE methodologies (Section 3.3).

### 3.1 DATA GENERATING PROCESS, POLICY, VALUE AND IS RATIO

**Data**. Assume the offline dataset $\mathcal{D}$ comprises multiple trajectories, each containing a sequence of state-action-reward triplets $(S_t, A_t, R_t)_{t \geq 1}$ following a finite MDP, denoted by $\mathcal{M} = \langle \mathcal{S}, \mathcal{A}, \mathcal{T}, \mathcal{R}, \rho_0, \gamma \rangle$. Here, $\mathcal{S}$ and $\mathcal{A}$ are the discrete state and action spaces, both with finite cardinalities, $\mathcal{T}$ and $\mathcal{R}$ are the state transition and reward functions, $\rho_0$ denotes the initial state distribution, and $\gamma \in (0, 1)$ is the discount factor. The data is generated as follows:

1. At the initial time, the state $S_1$ is generated according to $\rho_0$;
2. Subsequently, at each time $t$, the agent finds the environment in a specific state $S_t \in \mathcal{S}$ and selects an action $A_t \in \mathcal{A}$ according to a behavior policy $b$ such that $\mathbb{P}(A_t = a|S_t) = b(a|S_t)$;
3. The environment delivers an immediate reward $R_t$ with an expected value of $\mathcal{R}(A_t, S_t)$, and transits into the next state $S_{t+1} \overset{d}{\sim} \mathcal{T}(\bullet \mid A_t, S_t)$ according to the transition function $\mathcal{T}$.

Notice that both the reward and transition functions rely only on the current state-action pair $(S_t, A_t)$, independent of the past data history. This ensures that the data satisfies the Markov assumption.

**Policy and value**. Let $\pi$ denote a given target policy we wish to evaluate. We use $\mathbb{E}^\pi$ and $\mathbb{P}^\pi$ to denote the expectation and probability assuming the actions are chosen according to $\pi$ at each time. The regular $\mathbb{E}$ and $\mathbb{P}$ without superscript are taken with respect to the behavior policy $b$. Our objective lies in estimating the expected cumulative reward under $\pi$, denoted by $J(\pi) = \mathbb{E}^\pi \left[ \sum_{t=1}^{+\infty} \gamma^{t-1} R_t \right]$ using the offline dataset generated under a different policy $b$. Additionally, denote $V^\pi$ and $Q^\pi$ as the state value function and state-action value function (better known as the Q-function), namely,

$$V^\pi(s) = \mathbb{E}^\pi \left[ \sum_{t=1}^{+\infty} \gamma^{t-1} R_t | S_1 = s \right] \text{ and } Q^\pi(a, s) = \mathbb{E}^\pi \left[ \sum_{t=1}^{+\infty} \gamma^{t-1} R_t | S_1 = s, A_1 = a \right]. \quad (1)$$

These functions are pivotal in developing value-based estimators, as described in Method 1 of Section 3.3. Moreover, we use $\pi^*$ to denote the optimal policy that maximizes $J(\pi)$, i.e., $\pi^* \in \arg\max_\pi J(\pi)$, and write the optimal Q- and value functions $Q^{\pi^*}$, $V^{\pi^*}$ as $Q^*$, $V^*$ for brevity.

**IS ratio**. We also introduce the IS ratio $\rho^\pi(a, s) = \pi(a|s)/b(a|s)$, which quantifies the discrepancy between the target policy $\pi$ and the behavior policy $b$. Furthermore, define the MIS ratio

$$w^\pi(a, s) = (1 - \gamma) \sum_{t \geq 1} \frac{\gamma^{t-1} \mathbb{P}^\pi(S_t = s, A_t = a)}{\lim_{T \to \infty} \mathbb{P}(S_T = s, A_T = a)}. \quad (2)$$

Here, the numerator represents the discounted probability of visiting a given state-action pair under the target policy $\pi$, a crucial component in policy-based learning for estimating $\pi^*$ (Sutton et al., 1999; Schulman et al., 2015). The denominator corresponds to the limiting state-action distribution under the behavior policy. These ratios are fundamental in constructing IS estimators, as detailed in Methods 2 and 3 of Section 3.3.

### 3.2 STATE ABSTRACTIONS FOR POLICY LEARNING

Let $\mathcal{M} = \langle \mathcal{S}, \mathcal{A}, \mathcal{T}, \mathcal{R}, \rho_0, \gamma \rangle$ be the ground MDP. A state abstraction $\phi$ is a mapping from the state space $\mathcal{S}$ to certain abstract state space $\mathcal{X} = \{\phi(s) : s \in \mathcal{S}\}$. Below, we review some commonly studied definitions of state abstraction designed for learning the optimal policy $\pi^*$; see Jiang (2018).

**Definition 1 ($\pi^*$-irrelevance)** $\phi$ *is $\pi^*$-irrelevant if there exists an optimal policy $\pi^*$, such that for any $s^{(1)}, s^{(2)} \in \mathcal{S}$ whenever $\phi(s^{(1)}) = \phi(s^{(2)})$, we have $\pi^*(a|s^{(1)}) = \pi^*(a|s^{(2)})$ for any $a \in \mathcal{A}$.*

**Definition 2 ($Q^*$-irrelevance)** $\phi$ *is $Q^*$-irrelevant if for any $s^{(1)}, s^{(2)} \in \mathcal{S}$ whenever $\phi(s^{(1)}) = \phi(s^{(2)})$, the optimal Q-function satisfies $Q^*(a, s^{(1)}) = Q^*(a, s^{(2)})$ for any $a \in \mathcal{A}$.*

Definitions 1 and 2 are easy to understand, requiring the optimal policy/Q-function to depend on a state $s$ only through its abstraction $\phi(s)$. In practical terms, these definitions encourage the transformation of raw MDP data into a new sequence of state-action-reward triplets $(\phi(S), A, R)$ for policy learning. However, the transformed data may not necessarily satisfy the Markov assumption. This leads us to define the following model-irrelevance, which aims to preserve the MDP structure while ensuring $\pi^*$- and $Q^*$-irrelevance.

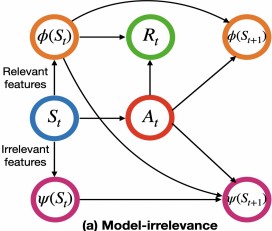 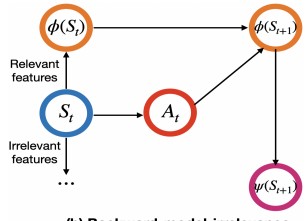

Figure 1: Illustrations of (a) model-irrelevance and (b) backward-model-irrelevance. $S_t$ is decomposed into the union of $\phi(S_t)$ (relevant features) and $\psi(S_t)$ (irrelevant features).

**Definition 3 (Model-irrelevance)** *$\phi$ is model-irrelevant if for any $s^{(1)}$, $s^{(2)} \in \mathcal{S}$ whenever $\phi(s^{(1)}) = \phi(s^{(2)})$, the following holds for any $a \in \mathcal{A}$, $s' \in \mathcal{S}$ and $x' \in \mathcal{X}$:*

$$\mathcal{R}(a, s^{(1)}) = \mathcal{R}(a, s^{(2)}) \quad and \quad \sum_{s' \in \phi^{-1}(x')} \mathcal{T}(s'|a, s^{(1)}) = \sum_{s' \in \phi^{-1}(x')} \mathcal{T}(s'|a, s^{(2)}). \tag{3}$$

The first condition in equation 3 corresponds to "reward-irrelevance" whereas the second condition represents "transition-irrelevance". Consequently, Definition 3 defines a "model-based" abstraction, in contrast to "model-free" abstractions considered in Definitions 1 and 2. Notice that the term $\sum_{s' \in \phi^{-1}(x')} \mathcal{T}(s'|a, s)$ – appearing in the second equation of equation 3 – represents the probability of transitioning to $\phi(S') = x'$ in the abstract state space. Thus, the second condition essentially requires the abstract next state $\phi(S')$ to be conditionally independent of $S$ given $A$ and $\phi(S)$. Assuming $S$ can be decomposed into the union of $\phi(S)$ and $\psi(S)$, which represent relevant features and irrelevant features, respectively. This condition implies that the evolution of those relevant features depends solely on themselves, independent of those irrelevant features. This ensures that the transformed data triplets $(\phi(S), A, R)$ remains an MDP. Meanwhile, the evolution of those irrelevant features may still depend on the relevant features; see Figure 1(a) for an illustration.

It is also known that model-irrelevance implies $Q^*$-irrelevance, which in turn implies $\pi^*$-irrelevance; see e.g., Theorem 2 in Li et al. (2006). Given that the transformed data remains an MDP under model-irrelevance, one can apply existing state-of-the-art RL algorithms to the abstract state space instead of the original ground space, leading to more effective learning of the optimal policy.

### 3.3 OPE METHODOLOGIES

We focus on four OPE methods, covering the three families of estimators introduced in Section 2. Each method employs a specific formula to identify $J(\pi)$, which we detail below. The first method is a popular value-based approach – the Q-function-based method. The second and third methods are the two major IS estimators: SIS and MIS. The fourth method is a semi-parametrically efficient doubly robust method, double RL (DRL), known for achieving the smallest possible MSE among a broad class of OPE estimators (Kallus & Uehara, 2020; 2022; Liao et al., 2022).

**Method 1 (Q-function-based method)**. For a given Q-function $Q$, define $f_1(Q)$ as the estimating function $\sum_{a \in \mathcal{A}} \pi(a|S_1)Q(a, S_1)$ with $S_1$ being the initial state. By equation 1 and the definition of $J(\pi)$, it is immediate to see that $J(\pi) = \mathbb{E}[f_1(Q^\pi)]$. This motivates the Q-function-based method which uses a plug-in estimator to approximate $\mathbb{E}[f_1(Q^\pi)]$ and estimate $J(\pi)$. In particular, $Q^\pi$ can be estimated by Q-learning type algorithms (e.g., fitted Q-evaluation, FQE, Le et al., 2019), and the expectation can be approximated based on the empirical initial state distribution.

**Method 2 (Sequential importance sampling)**. For a given IS ratio $\rho^\pi$, let $\rho^\pi_{1:t}$ denote the sequential IS ratio $\prod_{j=1}^{t} \rho^\pi(A_j, S_j)$. It follows from the change of measure theorem that the counterfactual reward $\mathbb{E}^\pi(R_t)$ is equivalent to $\mathbb{E}(\rho^\pi_{1:t}R_t)$ whose expectation is taken with respect to the offline data distribution. Assuming all trajectories in $\mathcal{D}$ terminate after a finite time $T$, this allows us to represent $J(\pi)$ by $\mathbb{E}[f_2(\rho^\pi)]$ where $f_2(\rho^\pi) = \sum_{t=1}^{T} \gamma^{t-1} \rho^\pi_{1:t} R_t$. SIS utilizes a plug-in estimator to first estimate $\rho^\pi$ (when the behavior policy is unknown), and then employs this estimator, along with the empirical data distribution, to approximate $\mathbb{E}[f_2(\rho^\pi)]$. However, a notable limitation of this estimator is its rapidly increasing variance due to the use of the SIS ratio $\rho^\pi_{1:t}$. Specifically, this

variance tends to grow exponentially with respect to $t$, a phenomenon often referred to as *the curse of horizon* (Liu et al., 2018).

**Method 3 (Marginalized importance sampling).** The MIS estimator is designed to overcome the limitations of the SIS estimator. It breaks the curse of horizon by taking the structure of the MDP model into account. As noted previously, under the Markov assumption, the reward depends only on the current state-action pair, rather than the entire history. This insight allows us to replace the SIS ratio with the MIS ratio, which depends solely on the current state-action pair. This modification considerably reduces variance because $w^\pi$ is no longer history-dependent. Assuming the data trajectory is stationary over time – that is, all state-action-reward $(S, A, R)$ triplets have the same distribution – it can be shown that $J(\pi) = \mathbb{E}[f_3(w^\pi)]$ where $f_3(w^\pi) = (1 - \gamma)^{-1} w^\pi(A, S) R$ for any triplet $(S, A, R)$. Both $w^\pi$ and the expectation can be effectively approximated using offline data.

**Method 4 (Double reinforcement learning).** DRL combines the Q-function-based method with MIS. Let $f_4(Q, w) = f_1(Q) + (1 - \gamma)^{-1} w(A, S)[R + \gamma \sum_a \pi(a|S')Q(a, S') - Q(A, S)]$, where $f_1$ is defined in Method 1 and $(S, A, R, S')$ denotes a state-action-reward-next-state tuple. Under the stationarity assumption, it can be shown that $J(\pi) = \mathbb{E}[f_4(Q, w)]$ when either $Q = Q^\pi$ or $w = w^\pi$ (Kallus & Uehara, 2022). DRL proposes to learn both $Q^\pi$ and $w^\pi$ from the data, employing these estimators to calculate $\mathbb{E}[f_4(Q, w)]$ and approximate the expectation with empirical data distribution. The resulting estimator benefits from double robustness: it is consistent when either $Q^\pi$ or $w^\pi$ is correctly specified.

# 4 PROPOSED STATE ABSTRACTIONS FOR POLICY EVALUATION

This section presents model-free (Section 4.1), model-based irrelevance conditions (Section 4.2) for OPE and analyzes the OPE estimators under these conditions (Lemma 1 & Theorem 1). Motivated by this analysis, we propose our iterative procedure (Section **??**) and study its property (Theorem 2).

Our theoretical analysis is concerned with the *Fisher consistency* of various OPE estimators, named after, Ronald Fisher, the founder of modern statistics. In particular, the Fisher consistency requires an estimator to be exactly equal to the ground truth given *infinite* samples. Specialized to our settings, it imposes two requirements:

(i) The identification formulas presented in Section 3.3 remain valid when replacing the oracle Q-function or (M)IS ratio with those projected into the proposed abstract state space;

(ii) The Q-function or (M)IS ratio defined on the abstract state space is identifiable.

Once the Fisher consistency is established, the estimator's *finite* sample properties can be readily obtained using existing techniques (see e.g., Uehara et al., 2021a). Therefore, for conciseness and to avoid redundancy, we chose not to present finite sample results in our theoretical analysis.

## 4.1 MODEL-FREE IRRELEVANCE CONDITIONS

We first introduce several model-free irrelevance conditions tailored for OPE.

**Definition 4 ($\pi$-irrelevance)** $\phi$ *is $\pi$-irrelevant if for any $s^{(1)}, s^{(2)} \in \mathcal{S}$ whenever $\phi(s^{(1)}) = \phi(s^{(2)})$, we have $\pi(a|s^{(1)}) = \pi(a|s^{(2)})$ for any $a \in \mathcal{A}$.*

**Definition 5 ($Q^\pi$-irrelevance)** $\phi$ *is $Q^\pi$-irrelevant if for any $s^{(1)}, s^{(2)} \in \mathcal{S}$ whenever $\phi(s^{(1)}) = \phi(s^{(2)})$, we have $Q^\pi(a, s^{(1)}) = Q^\pi(a, s^{(2)})$ for any $a \in \mathcal{A}$.*

Definitions 4 and 5 are adaptations of Definitions 1 and 2 designed for policy evaluation, with the optimal policy $\pi^*$ replaced by the target policy $\pi$. The following definitions are tailored for IS estimators (see Methods 2 and 3 in Section 3.3).

**Definition 6 ($\rho^\pi$-irrelevance)** $\phi$ *is $\rho^\pi$-irrelevant if for any $s^{(1)}, s^{(2)} \in \mathcal{S}$ whenever $\phi(s^{(1)}) = \phi(s^{(2)})$, we have $\rho^\pi(a, s^{(1)}) = \rho^\pi(a, s^{(2)})$ for any $a \in \mathcal{A}$.*

**Definition 7 ($w^\pi$-irrelevance)** $\phi$ *is $w^\pi$-irrelevant if for any $s^{(1)}, s^{(2)} \in \mathcal{S}$ whenever $\phi(s^{(1)}) = \phi(s^{(2)})$, we have $w^\pi(a, s^{(1)}) = w^\pi(a, s^{(2)})$ for any $a \in \mathcal{A}$.*

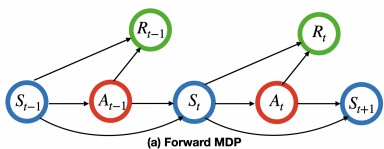 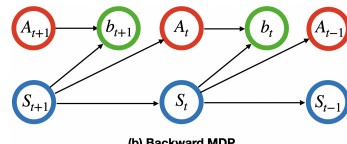

Figure 2: Illustrations of (a) the forward MDP model and (b) the backward MDP model. $b_t$ is a shorthand for $b(A_t|S_t)$ for any $t \geq 1$.

These irrelevance conditions encourage us to conduct OPE on the abstract state space to reduce sample complexity. Nevertheless, methods for deriving abstractions that satisfy these conditions (particularly $Q^\pi$- and $w^\pi$-irrelevance) remain unclear. Furthermore, the state-action-reward triplets transformed via these abstractions $(\phi(S), A, R)$ might not maintain the MDP structure. This complicates the process of learning $Q_\phi^\pi$ and $w_\phi^\pi$. These challenges motivate us to consider model-based irrelevance conditions introduced in the subsequent section.

### 4.2 MODEL-BASED IRRELEVANCE CONDITIONS

To begin with, we discuss two perspectives of the data generated within the MDP framework; see Figure 2 for a graphical illustration.

1. The first perspective is the traditional **forward MDP** model with all state-action-reward triplets sequenced by time index. This yields the model-based irrelevance condition defined in Definition 3. We will discuss the relationship between this condition and Definitions 5-7 below.
2. The second perspective offers a backward view by reversing the time order. Specifically, due to the symmetric nature of the Markov assumption — implying that if the future is independent of the past given the present, the past must also be independent of the future given the present — the reversed state-action pairs also maintain the Markov property. Leveraging this property, we define another **backward MDP**, which forms the basis for deriving model-based conditions for achieving $w^\pi$-irrelevance and motivates the subsequent iterative procedure. This development represents one of our main contributions.

**Forward MDP-based model-irrelevance**. We first discuss the connections between the model-irrelevance given in Definition 3 and the notions of $Q^\pi$-, $\rho^\pi$- and $w^\pi$-irrelevance, and introduce the following conditions to establish Fisher consistency. These conditions are mild and frequently imposed in the RL and OPE literature (see e.g., Thomas & Brunskill, 2016; Kallus & Zhou, 2018; Chen & Jiang, 2019; Fan et al., 2020; Cai et al., 2021; Shi et al., 2021; Kallus & Uehara, 2022).

**Assumption 1 (Boundedness)** *All immediate rewards are uniformly bounded.*

**Assumption 2 (Coverage)** *The denominator in equation 2 is strictly positive.*

**Assumption 3 (Stationary)** *The MDP $(S_t, A_t, R_t)_{t \geq 1}$ is stationary over time.*

The following lemma summarizes the findings. Results in the first two bullet points are based on those in the existing literature (see e.g., Li et al., 2006; Pavse & Hanna, 2023).

**Lemma 1 (OPE under model-irrelevance)** *Assume Assumptions 1–3 hold. Let $\phi$ denote a model-irrelevant abstraction. Suppose $\phi$ is additionally $\pi$-irrelevant. Then:*
- *$Q^\pi$-irrelevance & Fisher consistency of Q-function-based method: $\phi$ is also $Q^\pi$-irrelevant, and the corresponding Q-function-based method (Method 1) is thus Fisher consistent, i.e., the Q-function $Q_\phi^\pi$ defined on the abstract space is identifiable and satisfies $\mathbb{E}[f_1(Q^\pi)] = \mathbb{E}[f_1(Q_\phi^\pi)]$;*
- *Fisher consistency of MIS: While $\phi$ is not necessarily $w^\pi$-irrelevant, MIS (Method 3) is Fisher consistent when applied to the abstract state space, i.e., the MIS ratio $w_\phi^\pi$ defined on the abstract state space is identifiable and satisfies $\mathbb{E}[f_3(w^\pi)] = \mathbb{E}[f_3(w_\phi^\pi)]$;*
- *Fisher consistency of SIS: While $\phi$ is not necessarily $\rho^\pi$-irrelevant, SIS (Method 2) with a history-dependent IS ratio (detailed in the proof of Lemma 1 in Appendix D.1) is Fisher consistent when applied to the abstract space;*
- *Fisher consistency of DRL: DRL (Method 4) is Fisher consistent when applied to the abstract state space.*

The first bullet point establishes the link between model-irrelevance and $Q^\pi$-irrelevance and proves the Fisher consistency of the Q-function-based method when applied to the abstract state space. To satisfy $Q^\pi$-irrelevance, we need both model-irrelevance and $\pi$-irrelevance. In our implementation, we first adapt existing algorithms to train a model-irrelevant abstraction $\phi$, parameterized via deep neural networks. We next combine $\phi(s)$ with $\{\pi(a|s) : a \in \mathcal{A}\}$ to obtain a new abstraction $\phi_{for}(s)$. This augmentation ensures $\phi_{for}(s)$ is $\pi$-irrelevant, and hence $Q^\pi$-irrelevant. Refer to Appendix B.1 for the detailed procedures.

The last three bullet points validate the SIS, MIS and DRL estimators, despite $\phi$ being neither $w^\pi$-irrelevant nor $\rho^\pi$-irrelevant. By definition, $\rho^\pi$-irrelevance can be achieved by selecting state features that adequately predict the IS ratio. However, methods for constructing $w^\pi$-irrelevant abstractions remain less clear. In the following, we introduce a backward MDP model-based irrelevance condition that ensures $w^\pi$-irrelevance.

**Backward MDP-based model-irrelevance**. To illustrate the rationale behind the proposed model-based abstraction, we introduce the backward MDP model by reversing the time index. Under the (forward) MDP model assumption described in Section 3.1 and that the behavior policy $b$ is not history-dependent, actions and states following $S_t$ are independent of those occurred prior to the realization of $S_t$. Accordingly, $(S_{t-1}, A_{t-1})$ is conditionally independent of $\{(S_k, A_k)\}_{k>t}$ given $S_t$. Recall that $T$ corresponds to the termination time of trajectories in $\mathcal{D}$. We define a time-reversed process consisting of state-action-reward triplets $\{(S_t, A_t, b(A_t|S_t)) : t = T, \ldots, 1\}$. Its dynamics is described as follows (see also Figure 2(b) for the configuration):

- **State-action transition**: Due to the aforementioned Markov property, the transition of the past state $S_{t+1}$ in the reversed process (future state in the original process) into the current state $S_t$ is independent of the past action $A_{t+1}$ in the reversed process (future action in the original process) while the behavior policy that generates $A_t$ depends on both the current state $S_t$ and the past state $S_{t+1}$ in the reversed process. This yields the time-reversed state-action transition function $\mathbb{P}(A_t = a, S_t = s|S_{t+1})$.
- **Reward generation**: For each state-action pair $(S_t, A_t)$, we manually set the reward to the behavior policy $b(A_t|S_t)$, which plays a crucial role in constructing IS estimators.

Given this MDP, analogous to Definition 3, our objective is to identify a state abstraction that is crucial for predicting the reward (e.g., the behavior policy) and the reversed transition function. We provide the formal definition of the proposed backward MDP-based model-irrelevance (short for backward-model-irrelevance) below.

**Definition 8 (Backward-model-irrelevance)** $\phi$ *is backward-model-irrelevant if for any* $s^{(1)}, s^{(2)} \in \mathcal{S}$ *whenever* $\phi(s^{(1)}) = \phi(s^{(2)})$, *the followings hold for any* $a \in \mathcal{A}$ , $x \in \mathcal{X}$ *and* $t \in \mathbb{N}^+$:

$$(i) \; b(a|s^{(1)}) = b(a|s^{(2)}); \tag{4}$$

$$(ii) \sum_{s \in \phi^{-1}(x)} \mathbb{P}(A_t = a, S_t = s|S_{t+1} = s^{(1)}) = \sum_{s \in \phi^{-1}(x)} \mathbb{P}(A_t = a, S_t = s|S_{t+1} = s^{(2)}). \tag{5}$$

The conditions of backward-model-irrelevance are similar to those specified for model-irrelevance outlined in Definition 3. Equation 4 essentially requires behavior-policy-irrelevance, or reward-irrelevance in the backward MDP. Equation 5 corresponds to the "backward-transition-irrelevance", and is equivalent to the conditional independence assumption between the pair $(A_t, \phi(S_t))$ and $S_{t+1}$ given $\phi(S_{t+1})$. As previously assumed, $S_t$ can be decomposed into the union of relevant features $\phi(S_t)$ and irrelevant features $\psi(S_t)$ (see Figure 1), leading to the following factorization:

$$\mathbb{P}(S_{t+1} = s'|A_t, \phi(S_t)) = \mathbb{P}(\psi(S_{t+1}) = \psi(s')|\phi(S_{t+1}) = \phi(s'))\mathbb{P}(\phi(S_{t+1}) = \phi(s')|A_t, \phi(S_t)).$$

This indicates a two-step transition in the forward model: initially from $(\phi(S_t), A_t)$ to $\phi(S_{t+1})$, and then from $\phi(S_{t+1})$ to $\psi(S_{t+1})$. Importantly, the generation of $\psi(S_{t+1})$ in the second step is conditionally independent of $A_t$ and $\phi(S_t)$. Consequently, $\phi$ extracts state representations that are influenced either by past actions or past relevant features; see Figure 1(b) for an illustration. Combined with $\pi$-irrelevance and behavior-policy-irrelevance, this ensures that all information contained within the historical IS ratios $\{\rho^\pi(A_k, S_k)\}_{k<t}$ can be effectively summarized using a single $A_{t-1}$ and the abstract state $\phi(S_{t-1})$, thus achieving $w^\pi$-irrelevance (see Theorem 1 below).

**Theorem 1 (OPE under backward-model-irrelevance)** *Assume Assumptions 1–3 hold, and $\phi$ is both backward-model-irrelevant and $\pi$-irrelevant.*

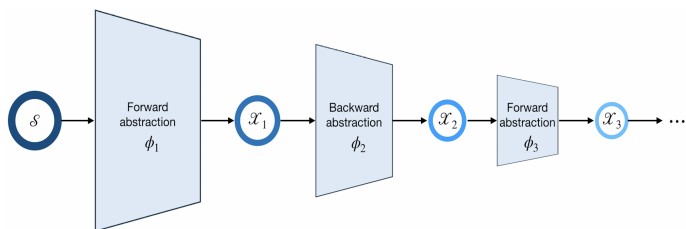

Figure 3: Illustrations of the iterative procedure.

- *Then $\phi$ is both $\rho^\pi$-irrelevant and $w^\pi$-irrelevant.*
- *Additionally, all the four OPE methods (i.e., Q-function-based, SIS, MIS, DRL) are Fisher consistent when applied to the abstract state space.*

Theorem 1 validates the four OPE methods when applied to the abstract state space. To conclude this section, we draw a connection between the proposed backward-model-irrelevant abstraction for OPE and the Markov state abstraction (MSA) developed by Allen et al. (2021) for policy learning. MSA impose two conditions: (i) inverse-model-irrelevance, which requires $A_t$ to be conditionally independent of $S_t$ and $S_{t+1}$ given $\phi(S_t)$ and $\phi(S_{t+1})$; (ii) density-ratio-irrelevance, which requires $\phi(S_t)$ to be conditionally independent of $S_{t+1}$ given $\phi(S_{t+1})$. For effective policy learning, MSA requires both conditions to hold in data generating processes following a diverse range of behavior policies. When restricting them to one behavior policy, the two conditions are closely related to our backward-model-irrelevance. In particular, they imply our proposed backward-transition-irrelevance condition in equation 5 whereas backward-transition-irrelevance in turn yields their density-ratio-irrelevance. This allows us to adapt their algorithm to train state abstractions that satisfy our proposed backward-model-irrelevance; see Appendix B.2 for details.

Finally, it is worth noting that both the proposed backward-model-irrelevant abstraction and MSA require the behavior policy to be Markov, independent of the past observations.The following section will relax this condition and extend our proposal to accommodate history-dependent behavior policies.

### 4.3 ITERATIVE PROCEDURE FOR DEEP STATE ABSTRACTION

To summarize, we have reviewed the model-irrelevance condition and proposed a new backward-model-irrelevance condition. Both lead to Fisher consistent OPE estimators when confined to the corresponding abstract state spaces. This motivates us to combine the two procedures for a more condensed state abstraction, resulting in the following iterative algorithm (see Figure 3 for a visualization):

1. **Forward abstraction**: learn an abstraction $\phi_1$ from the ground state space $\mathcal{S} = \mathcal{X}_0$ to $\mathcal{X}_1$ using the data triplets $(S, A, R)$ that is both (forward)-model-irrelevant and $\pi$-irrelevant.
2. **Backward abstraction**: Learn an abstraction $\phi_2$ from the abstract state space $\mathcal{X}_1$ to $\mathcal{X}_2$ using the data triplets $(\phi_1(S), A, R)$ that is both backward-model-irrelevant and $\pi$-irrelevant.
3. **Iterate** the two steps to compute the final abstraction $\phi_K \circ \cdots \circ \phi_2 \circ \phi_1$ from the ground space $\mathcal{S}$ to $\mathcal{X}_K$ where $K$ denotes the number of iterations and $\circ$ denotes the function composition operator.

In particular, our approach alternates between forward and backward abstraction on the state space obtained from the previous iteration. Each iteration guarantees that the cardinality of the state space does not increase, effectively maintaining or reducing complexity. Consequently, such an iterative procedure progressively reducing state cardinality, which ultimately yields a deeply-abstracted state. We thus refer to our approach as deep state abstraction (DSA).

To elaborate the usefulness of DSA in reducing state cardinality, we analyze two examples: a bandit example and an MDP example. In both examples, we focus on a specific type of state abstraction known as variable selection, which selects a sub-vector from the original state. Additionally, we focus on the class of state-agnostic target policies where $\pi$ is independent of the states. This type of policy is prevalent in causal inference and A/B testing, where the objective is to learn the global

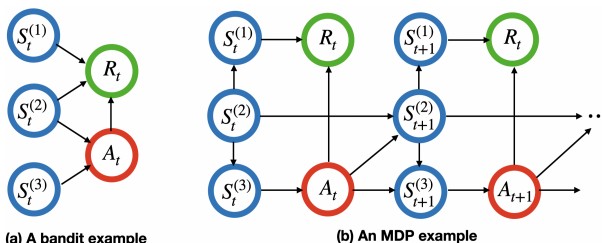

Figure 4: Two illustrative examples.

treatment effect of applying either a new or old policy consistently over time (see e.g., Hu & Wager, 2022; Leung, 2022; Bojinov et al., 2023; Shi et al., 2023; Xiong et al., 2024).

**Example I (A bandit example)**: To illustrate the main idea, we start by considering the contextual bandit setting (CB, Dudík et al., 2014), which can be regarded as a special MDP with independent state transitions. Under this setting, the states are i.i.d. generated, and model-irrelevance is reduced to reward-irrelevance whereas the proposed backward-model-irrelevance simplifies to behavior-policy-irrelevance. When specialized to variable selection in CB, our proposal is reduced to the iterative confounder selection algorithm in causal inference (see e.g., Guo et al., 2022); see also the review of confounder selection in Appendix A. We assume the states can be divided into three independent groups, denoted by $S_t^{(1)}$, $S_t^{(2)}$ and $S_t^{(3)}$, respectively. Each group influences the system differently: $S_t^{(1)}$ affects only the reward, $S_t^{(2)}$ impacts both the action and the reward, and $S_t^{(3)}$ solely influences the action; see Figure 4(a) for an illustration. As formally proven in Lemma D.3 (see Appendix D.5):

- The forward abstraction selects the first two groups $S_t^{(1)}$ and $S_t^{(2)}$;
- The proposed backward abstraction selects the last two groups $S_t^{(2)}$ and $S_t^{(3)}$;
- The proposed DSA selects their intersection $S_t^{(2)}$ and converges in two steps, resulting in a smaller subset of variables compared to the two non-iterative procedures.

**Example II (An MDP example)**: We next consider an MDP with three groups of states, depicted in Figure 4(b). Key observations from this example are as follows: (i) The reward depends on the state only through the first group of variables; (ii) The evolution of the state depends only on the second group. Specifically, the second group evolves first at each time step and subsequently influences the rest two groups; (iii) The behavior policy depends only on the last group; (iv) The second group is directly influenced by the previous action. Based on these observations, we show have that:

- According to (i), selecting the first group of variables achieves reward-irrelevance.
- According to (iii), selecting the last group of variables achieves behavior-policy-irrelevance.
- According to (ii) and (iv), selecting the second group achieves both transition-irrelevance (see the second equation in 3) and backward-transition-irrelevance (see equation 5).

Consequently, the forward and backward abstractions select the first two and last two groups of variables, respectively. The iterative procedure again selects their intersection $S_t^{(2)}$ and converges in two iterations, leading to in a smaller state space. The reader is referred to Appendix D.5 for formal justifications.

In both examples, we have demonstrated the advantage of DSA in reducing state space cardinality over non-iterative procedures. However, in more general scenarios, two challenges arise from the iterative nature of DSA: (i) After forward abstraction, the behavior policy when restricted to the abstract space can be history-dependent. This would invalidate the subsequent backward abstraction for achieving $w^\pi$-irrelevance. (ii) After backward abstraction, the Markov assumption might be violated. This would invalidate the subsequent forward abstraction for consistent OPE.

To address both challenges, we modify the proposed backward-model-irrelevance by employing a history-dependent behavior policy $b_t(a_t|s_t, a_{t-1}, s_{t-1}, \cdots, s_1) = \mathbb{P}(A_t = a_t | S_t = s_t, A_{t-1} = a_{t-1}, S_{t-1} = s_{t-1}, \cdots, S_1 = s_1)$ in equation 4. Specifically, we require that for any two state sequences $\{s_\ell^{(1)}\}_\ell$, $\{s_\ell^{(2)}\}_\ell$ such that $\phi(s_\ell^{(1)}) = \phi(s_\ell^{(2)})$, any action sequence $\{a_\ell\}_\ell$ and any time index $t \geq 1$,

$$b_t(a_t|s_t^{(1)}, a_{t-1}, s_{t-1}^{(1)}, \cdots, s_1^{(1)}) = b_t(a_t|s_t^{(2)}, a_{t-1}, s_{t-1}^{(2)}, \cdots, s_1^{(2)}). \tag{6}$$

The following theorem validates the abstraction produced by the proposed DSA at *any* iteration.

**Theorem 2 (OPE under the iterative procedure)** *Assume Assumptions 1–3 hold. With the refined backward-model-irrelevance, the four OPE methods are Fisher consistent when applied to the abstract state space produced by the proposed DSA, regardless of the number of iterations conducted.*

Finally, we note that the initialization of the iterative procedure doesn't necessarily have to begin with forward abstraction; backward abstraction can serve as the starting point as well. In our experiments, both starting points have their merits, with their effectiveness varying depending on the environment. However, the overall differences in results are small.

## 5 NUMERICAL EXPERIMENTS

We investigate the finite sample performance of our proposal in this section and detail its implementation in Appendix B.

**Comparisons**. We compare the proposed deep state abstraction (denoted by 'DSA') against single-iteration forward ('forward'), backward ('backward') abstractions, the Markov state abstraction (Allen et al., 2021) ('MSA') and a reconstruction-based abstraction (Lange & Riedmiller, 2010) ('auto-encoder'). For fairer comparison, each abstraction's performance is tested by applying a base FQE algorithm (Le et al., 2019) applied to the abstract state space. We also report the performance of FQE applied to the unabstracted, ground state space ('FQE').

**Environment**. We consider the "LunarLander-v2" environment in this section, with an original state dimension of 8. We manually include 292 irrelevant variables in the state, leading to a challenging 300-dimensional system. Refer to Appendix C for more details about the environment.

**Results**. We report MSEs and absolute biases of different post-abstraction-OPE estimators and those of the baseline FQE estimator without abstraction in Figure 5. It can be seen that the proposed DSA method outperforms other baseline methods, with the smallest relative MSE and absolute bias in most cases. To conclude, our analysis answers the following questions:

1. **Is state abstraction useful for OPE**? Both figures show that the baseline FQE applied to the ground state space performs the worst among all cases. This comparison reveals the usefulness of state abstractions for OPE.
2. **Is the deep/iterative procedure more effective compared to single-iteration procedure?** Notice that 'Markov' and 'auto-encoder' are types of model-irrelevant abstractions. The comparisons against these abstractions as well as 'forward' and 'backward' demonstrate the advantages of DSA over single-iteration procedures.

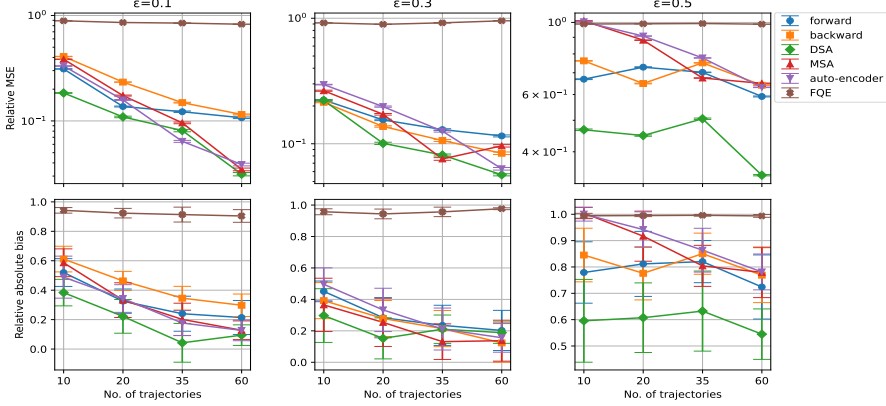

Figure 5: Relative MSEs and absolute biases of FQE estimators when applied to ground and abstract state spaces with various abstractions. The behavior policy is $\epsilon$-greedy with respect to the target policy, with $\epsilon = 0.1, 0.3, 0.5$ from left to right.

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

## APPENDIX

This appendix is structured as follows: Section A introduces additional related works on confounder selection in causal inference. The implementation details of the proposed state abstraction are discussed in Section B. Additional information concerning the environment and computing resources utilized is presented in Section C. All technical proofs can be found in Section D.

## A CONFOUNDER SELECTION IN CAUSAL INFERENCE

Broadly speaking, confounding refers to the problem that even if two variables are not causes of each other, they may exhibit statistical association due to common causes. Controlling for confounding is a central problem in the design of observational studies, and many criteria for confounder selection have been proposed in the literature. A commonly adopted criterion is the "common cause heuristic", where the user only controls for covariates that are related to both the treatment and the outcome (Glymour et al., 2008; Austin, 2011; Shortreed & Ertefaie, 2017; Koch et al., 2020). Another widely used criterion is to simply use all covariates that are observed before the treatment in time (Rubin, 2009; Hernán & Robins, 2010; 2016). However, both of these approaches are not guaranteed to find a set of covariates that are sufficient to control for confounding. From a graphical perspective, confounder selection is essentially about finding a set of covariates that block all "back-door" paths (Pearl, 2009), but this requires full structural knowledge about the causal relationship between the variables which is often not possible. This motivated some methods that only require partial structural knowledge (Vander Weele & Shpitser, 2011; VanderWeele, 2019; Guo & Zhao, 2023). All the aforementioned methods need substantive knowledge about the treatment, outcome, and covariates.

Other methods use statistical tests (usually of conditional independence) to trim a set of covariates that are assumed to control for confounding (Robins, 1997; Greenland et al., 1999; Hernán & Robins, 2010; De Luna et al., 2011; Belloni et al., 2014; Persson et al., 2017). The reader is referred to Guo et al. (2022) for a recent survey of objectives and approaches for confounder selection.

Confounder selection can be considered as a special example of our problem under certain conditions: (i) The state transition is independent, effectively transforming the MDP into a contextual bandit; (ii) The action space is binary, with the target policy consistently assigning either action 0 or action 1, aimed at assessing the average treatment effect; (iii) State abstractions are confined to variable selections. While our proposed iterative procedure shares similar spirits with the aforementioned algorithms, it addresses a more complex problem involving state transitions. Additionally, our focus is on abstraction that facilitates the engineering of new feature vectors, rather than merely selecting a subset of existing ones.

# B IMPLEMENTATION DETAILS

In this section, we present implementation details for forward abstraction (Section B.1) and backward abstraction (Section B.2).

## B.1 IMPLEMENTATION DETAILS FOR FORWARD ABSTRACTION

We provide details for implementing the proposed forward abstraction in this subsection. We use deep neural networks to parameterize the forward abstraction and estimate the parameters by minimzing the following loss function:

$$\alpha_1 \mathcal{L}_r + \beta_1 \mathcal{L}_\mathcal{T} + \delta_1 \mathcal{L}_Q + \lambda_1 \mathcal{L}_{penalty}, \tag{B.1}$$

where $\mathcal{L}_r$, $\mathcal{L}_\mathcal{T}$ and $\mathcal{L}_Q$ are the loss functions detailed below, $\mathcal{L}_{penalty}$ is a penalty term, and $\alpha_1, \beta_1, \delta_1, \lambda_1$ are positive constant hyper-parameters whose values are reported in Table B.1.

By definition, the forward abstraction is required to achieve both model-irrelevance and $\pi$-irrelevance. As discussed in Section 4.2, our approach is to learn a model-irrelevant abstraction, denoted as $\phi$, and then concatenate it with $\{\pi(a|\bullet) : a \in \mathcal{A}\}$. We denote the concatenated abstraction by $\phi_{for}$.

We next detail the loss functions and the penalty term. The first two losses $\mathcal{L}_r$ and $\mathcal{L}_\mathcal{T}$ are to ensure reward-irrelevance and transition-irrelevance, respectively,

$$\mathcal{L}_r = \frac{1}{|\mathcal{D}|} \sum_{(S,A,R)\in\mathcal{D}} \left[ R - \mathcal{R}_\phi\big(A, \phi(S)\big) \right]^2, \ \ \mathcal{L}_\mathcal{T} = \frac{1}{|\mathcal{D}|} \sum_{(S,A,S')\in\mathcal{D}} \left\| \mathcal{T}_\phi\big(A, \phi(S)\big) - \phi(S') \right\|_2^2,$$

where $\mathcal{R}_{\phi_0}$ and $\mathcal{T}_{\phi_0}$ are the estimated reward and transition functions applied to the abstract state space parameterized by deep neural networks as well, and $|\mathcal{D}|$ is the cardinality of the dataset $\mathcal{D}$.

The inclusion of the third loss function, $\mathcal{L}_Q$, is motivated by the demonstrated benefits of utilizing model-free objectives to guide the training of state abstractions in policy learning François-Lavet et al. (2019).

Given our interest in OPE, we integrate the following FQE loss into the objective function,

$$\mathcal{L}_Q = \frac{1}{|\mathcal{D}|} \sum_{(S,A,R,S')\in\mathcal{D}} \left[ R + \gamma \sum_{a\in\mathcal{A}} \pi(a|S')Q^-\big(\phi_{for}(S'), a\big) - Q\big(\phi_{for}(S), A\big) \right]^2,$$

where $Q^-$ and $Q$ represent the estimated $Q^\pi_{\phi_{for}}$ function applied to the abstract state space during the previous and current iterations, respectively.

The above objectives allow us to effectively train forward abstractions. However, a potential concern is that the resulting abstraction and transition can collapse to some constant $x_0$ such that

$\phi_{for}(S) \to x_0, \ \forall S \in \mathcal{S}$. To address this limitation, we include the following penalty function of two randomly drawn states to promote diversity in the abstractions:

$$\mathcal{L}_c = \frac{1}{|\mathcal{D}|(|\mathcal{D}|-1)} \sum_{S,\tilde{S} \in \mathcal{D}, S \neq \tilde{S}} \exp(-C_0 \|\widehat{\phi}(S) - \widehat{\phi}(\tilde{S})\|_2)$$

for some positive scaling constant $C_0$, and $\widehat{\phi}(s)$ is the estimated abstract state from transition function. $\widehat{\phi}(\tilde{s})$ can be achieved by shuffling $\widehat{\phi}(s')$ from pairs $(s, s')$ in the batch. Additionally, we add another penalty to penalize consecutive abstract states for being more than some predefined distance $d_0$ away from each other,

$$\mathcal{L}_s = \frac{1}{|\mathcal{D}|} \sum_{(S,S') \in \mathcal{D}} C_1 [\|\phi_{for}(S) - \phi_{for}(S')\|_2 - d_0]^2,$$

for some positive constant $C_1$. These components combine into the final penalty function:

$$\mathcal{L}_{penalty} = \mathcal{L}_s + \mathcal{L}_c.$$

The forward model architecture is as follow:

```
  Forward_model(
 (encoder): Encoder_linear(
   (activation): ReLU()
   (encoder_net): Sequential(
     (0): Linear(in_features=300, out_features=64, bias=True)
     (1): ReLU()
     (2): Linear(in_features=64, out_features=64, bias=True)
     (3): ReLU()
     (4): Dropout(p=0.2, inplace=False)
     (5): Linear(in_features=64, out_features=64, bias=True)
     (6): ReLU()
     (7): Dropout(p=0.2, inplace=False)
     (8): Linear(in_features=64, out_features=100, bias=True)
   )
 )
 (transition): Transition(
   (activation): ReLU()
   (T_net): Sequential(
     (0): Linear(in_features=100, out_features=64, bias=True)
     (1): ReLU()
     (2): Linear(in_features=64, out_features=64, bias=True)
     (3): ReLU()
     (4): Dropout(p=0.2, inplace=False)
     (5): Linear(in_features=64, out_features=64, bias=True)
   )
   (lstm): LSTMCell(64, 128)
   (tanh): Tanh()
 )
 (reward): Reward(
   (activation): ReLU()
   (reward_net): Sequential(
     (0): Linear(in_features=100, out_features=64, bias=True)
     (1): ReLU()
     (2): Linear(in_features=64, out_features=64, bias=True)
     (3): ReLU()
     (4): Dropout(p=0.2, inplace=False)
     (5): Linear(in_features=64, out_features=64, bias=True)
     (6): ReLU()
     (7): Dropout(p=0.2, inplace=False)
```

Table B.1: Hyper-parameters information. $m$ is the input feature dimension, and $**$ means no value.

| Hyper-parameters | Values | Hyper-parameters | Values |
|---|---|---|---|
| $\alpha_1$ | 1 | $\alpha_2$ | 1 |
| $\beta_1$ | 1 | $\beta_2$ | 1 |
| $\gamma_1$ | 1 | $\gamma_2$ | 1 |
| $\lambda_1$ | $\min(1, \frac{20}{m})$ | $\lambda_2$ | $\min(1, \frac{20}{m})$ |
| $C_0$ | 1 | $C_0$ | $**$ |
| $C_1$ | 1 | $C_1$ | 1 |
| $d_0$ | $0.15m$ | $d_0$ | $0.15m$ |

```
    (8): Linear(in_features=64, out_features=64, bias=True)
    (9): ReLU()
    (10): Dropout(p=0.2, inplace=False)
    (11): Linear(in_features=64, out_features=64, bias=True)
    (12): ReLU()
    (13): Dropout(p=0.2, inplace=False)
    (14): Linear(in_features=64, out_features=2, bias=True)
  )
)
(FQE): FQE(
  (activation): ReLU()
  (action_net): Sequential(
    (0): Linear(in_features=1, out_features=16, bias=True)
    (1): ReLU()
    (2): Linear(in_features=16, out_features=100, bias=True)
  )
  (xa_net): Linear(in_features=200, out_features=100, bias=True)
  (FQE_net): Sequential(
    (0): Linear(in_features=100, out_features=64, bias=True)
    (1): ReLU()
    (2): Linear(in_features=64, out_features=64, bias=True)
    (3): ReLU()
    (4): Dropout(p=0.2, inplace=False)
    (5): Linear(in_features=64, out_features=64, bias=True)
    (6): ReLU()
    (7): Dropout(p=0.2, inplace=False)
    (8): Linear(in_features=64, out_features=2, bias=True)
  )
)
)
```

## B.2 IMPLEMENTATION DETAILS FOR BACKWARD ABSTRACTION

We provide details for implementing the proposed backward abstraction in this subsection. Similar to Section B.1, we use deep neural networks to parameterize the abstraction $\phi_{back}$ and estimate the parameters by solving the following loss function,

$$\alpha_2 \mathcal{L}_b + \beta_2 \mathcal{L}_{ratio} + \delta_2 \mathcal{L}_{inv} + \lambda_2 \mathcal{L}_s,$$

where $\alpha_2, \beta_2, \delta_2, \lambda_2$ are positive hyper-parameters specified in Table B.1.

Recall that backward-model-irrelevance requires both $\rho^\pi$-irrelevance (Definition 6) and equation 5. To enforce $\rho^\pi$-irrelevance, we first introduce behavior-irrelevance, this can be achieved by minimizing

the following cross-entropy loss for behavior:

$$\mathcal{L}_b = -\frac{1}{|\mathcal{D}|} \sum_{(S,A)\in\mathcal{D}} \log b\big(A = a|S\big)$$

and followed by concatenating with $\{\pi(a|\bullet) : a \in \mathcal{A}\}$, which ensures $\pi$-irrelevance. Note that in deeply-abstracted procedure, we replace the behavior loss by

$$\mathcal{L}_b = -\frac{1}{|\mathcal{D}|} \sum_{(S_t,A_t)\in\mathcal{D}} \log b\big(A_t = a_t|\phi_1(S_t), \{A_{t-k}, \phi_1(S_{t-k})\}_{k=1,2,\ldots}\big)$$

which incorporates history information as mentioned in 4.3. In practice, we do not use all the history information for behaviour policy, instead we use the history up to past two steps: $b(A_t|\phi_1(S_t), A_{t-1}, \phi_1(S_{t-1}), A_{t-2}, \phi_1(S_{t-2}))$.

As commented in Section 4.2, equation 5 holds by satisfying the conditional independence assumption between $(A_t, \phi(S_t))$ and $S_{t+1}$ given $\phi(S_{t+1})$. By Bayesian formula, we can show that it is satisfied by the inverse-model-irrelevance and density-ratio-irrelevance when setting the learning policy $\pi$ to $b$. This motivates us to leverage the two objectives $\mathcal{L}_{inv}$ and $\mathcal{L}_{ratio}$ used by Allen et al. (2021) for training MSA. More details regarding these losses can be found in Section 5 of Allen et al. (2021). Note that to obtain non-sequential states $(s, \tilde{s})$ used in $L_{ratio}$, we flip $s'$ in the pairs $(s, s')$ in each batch instead of shuffling.

Finally, $\mathcal{L}_s$ corresponds to the smoothness penalty introduced in Section B.1. The backward model architecture is:

```
    Backward_model(
  (encoder): Encoder_linear(
    (activation): ReLU()
    (encoder_net): Sequential(
      (0): Linear(in_features=100, out_features=64, bias=True)
      (1): ReLU()
      (2): Linear(in_features=64, out_features=64, bias=True)
      (3): ReLU()
      (4): Dropout(p=0.2, inplace=False)
      (5): Linear(in_features=64, out_features=64, bias=True)
      (6): ReLU()
      (7): Dropout(p=0.2, inplace=False)
      (8): Linear(in_features=64, out_features=6, bias=True)
    )
  )
  (inverse): Inverse(
    (activation): ReLU()
    (inverse_net): Sequential(
      (0): Linear(in_features=12, out_features=64, bias=True)
      (1): ReLU()
      (2): Linear(in_features=64, out_features=64, bias=True)
      (3): ReLU()
      (4): Dropout(p=0.3, inplace=False)
      (5): Linear(in_features=64, out_features=64, bias=True)
      (6): ReLU()
      (7): Dropout(p=0.3, inplace=False)
      (8): Linear(in_features=64, out_features=64, bias=True)
      (9): ReLU()
      (10): Dropout(p=0.3, inplace=False)
      (11): Linear(in_features=64, out_features=64, bias=True)
      (12): ReLU()
      (13): Dropout(p=0.3, inplace=False)
      (14): Linear(in_features=64, out_features=1, bias=True)
```

```
        )
      )
    (density): Density(
      (activation): ReLU()
      (density_net): Sequential(
        (0): Linear(in_features=12, out_features=64, bias=True)
        (1): ReLU()
        (2): Linear(in_features=64, out_features=64, bias=True)
        (3): ReLU()
        (4): Dropout(p=0.3, inplace=False)
        (5): Linear(in_features=64, out_features=64, bias=True)
        (6): ReLU()
        (7): Dropout(p=0.3, inplace=False)
        (8): Linear(in_features=64, out_features=64, bias=True)
        (9): ReLU()
        (10): Dropout(p=0.3, inplace=False)
        (11): Linear(in_features=64, out_features=64, bias=True)
        (12): ReLU()
        (13): Dropout(p=0.3, inplace=False)
        (14): Linear(in_features=64, out_features=1, bias=True)
      )
    )
    (rho): Rho(
      (activation): ReLU()
      (rho_net): Sequential(
        (0): Linear(in_features=6, out_features=64, bias=True)
        (1): ReLU()
        (2): Linear(in_features=64, out_features=64, bias=True)
        (3): ReLU()
        (4): Dropout(p=0.3, inplace=False)
        (5): Linear(in_features=64, out_features=64, bias=True)
        (6): ReLU()
        (7): Dropout(p=0.3, inplace=False)
        (8): Linear(in_features=64, out_features=2, bias=True)
      )
    )
  )
)
```

## C    ADDITIONAL EXPERIMENTAL DETAILS

### C.1    REPRODUCIBILITY

We release our code and data on the website at
`https://anonymous.4open.science/r/state-abstraction-588A/README.md`
The hyper-parameters to train the proposed forward and backward abstractions can be found in
Table B.1.

### C.2    EXPERIMENTAL SETTINGS AND ADDITIONAL RESULTS

For the environment LunarLander, we use Adam Kingma & Ba (2014) optimizer with learning rate
0.003. Model architectures and hyper-parameters are outlined in B. When conducting OPE, the FQE
network has 3 hidden layers with 64 nodes per hidden layer for abstraction methods, and is equipped
with 5 hidden layers with 128 nodes per hidden layer for non-abstracted observations (shown as
'FQE' in the plot).

**Data generating processes**

We similarly insert 292 irrelevant auto-regressive features in the state:

$$\mathbb{P}(S_{t+1,j}|S_t, A_t) = \mathbb{P}(S_{t+1,j}|S_{t,j}), \quad j = 9, \ldots, 300.$$

The number of trajectories $n$ in the offline dataset is chosen from $\{10, 20, 35, 60\}$, where trajectory length differs significantly in this environment. Some lengthy episodes can have length larger than 100000 while short episodes have fewer than 100 decision points. When trained and evaluated on the short episodes, OPE methods will fail due to huge distributional drift. We therefore truncate the episode length at 1000 if it exceeds, define it as long episode and those fewer than 1000 as short episodes. When generating trajectories, we use a long-short combination for each size: $\{10 = 7_{long} + 3_{short}, 20 = 14_{long} + 6_{short}, 35 = 25_{long} + 10_{short}, 60 = 45_{long} + 15_{short}\}$. The target policy is an estimated optimal policy pre-trained by an DQN agent whereas the behavior policy again $\epsilon$-greedy to the target policy with $\epsilon \in \{0.1, 0.3, 0.5\}$. Results are averaged over 20 runs for each $(n, \epsilon)$ pair and are reported in Figure 5

**Model parameters**

For forward and backward models, we abstract the original state dimension from $300 \to 10$, and for DSA method we reduce dimensions from $300 \to 100 \to 50 \to 20 \to 6$, by [backward, forward, backward, forward] order.

**Pre-trained agent**

We pre-train an agent by using DQN as our target policy. The agent is trained until there exists an episode that has accumulative discounted rewards exceeding 200 with discounted rate $\gamma = 0.99$. We evaluated oracle value (61.7) of the optimized agent by Monte Carlo method with the same discounted rate. The agent model architecture is as follow:

```
  DQN(
(fc1): Linear(in_features=8, out_features=64, bias=True)
(fc2): Linear(in_features=64, out_features=64, bias=True)
(fc3): Linear(in_features=64, out_features=4, bias=True)
)
```

## C.3 LICENCES FOR EXISTING ASSETS

We consider the environment from OpenAI Gym (Brockman et al., 2016) "LunarLander-v2" with the MIT License and Copyright (c) 2016 OpenAI (https://openai.com).

## C.4 COMPUTING RESOURCES

To build Figure 5, we trained 3 abstraction methods and one non-abstraction method on 4 different sizes of data, each with 20 runs, under 3 $\epsilon$ values. In average, each run takes approximately 8 minutes for four methods on an E2-series CPU with 64GB memory on GCP. It takes about 32 compute hours to complete all the experiments in the figure.

## D TECHNICAL PROOFS

We provide the detailed proofs of our theorems (Lemma 1, Theorems 1 & 2) in this section.

**Notations**. For events or random variables $A, B, C$, $A \perp\!\!\!\perp B$ means the independence between $A$ and $B$ whereas $A \perp\!\!\!\perp B|C$ means the conditional independence between $A$ and $B$ given $C$.

**An auxiliary lemma**. To begin with, we introduce the following lemma which validates the unbiasedness of various OPE estimators under the model-free irrelevance conditions in Definitions 5 – 7, whose proof is given in Section D.1.

**Lemma D.1 (Unbiasedness under model-free irrelevance conditions)** *Under $Q^\pi$-, $\rho^\pi$- or $w^\pi$-irrelevance, the corresponding methods are unbiased when applied to the abstract state space, assuming the oracle Q-function or (M)IS ratio is identifiable from the data:*

- *Under $Q^\pi$-irrelevance, Q-function-based method (Method 1) remains unbiased, i.e., the Q-function $Q_\phi^\pi$ defined on the abstract space satisfies $\mathbb{E}[f_1(Q^\pi)] = \mathbb{E}[f_1(Q_\phi^\pi)]$;*
- *Under $\rho^\pi$-irrelevance, SIS (Method 2) remains unbiased, i.e., the IS ratio $\rho_\phi^\pi$ defined on the abstract state space satisfies $\mathbb{E}[f_2(\rho^\pi)] = \mathbb{E}[f_2(\rho_\phi^\pi)]$;*
- *Under $w^\pi$-irrelevance, MIS (Method 3) remains unbiased, i.e., the MIS ratio $w_\phi^\pi$ defined on the abstract state space satisfies $\mathbb{E}[f_3(w^\pi)] = \mathbb{E}[f_3(w_\phi^\pi)]$.*

*Moreover, when $\phi$ satisfies either $Q^\pi$-irrelevance or $w^\pi$-irrelevance, DRL (Method 4) remains unbiased, i.e., $Q_\phi^\pi$ and $w_\phi^\pi$ defined on the abstract state space satisfy $\mathbb{E}[f_4(Q^\pi, w^\pi)] = \mathbb{E}[f_4(Q_\phi^\pi, w_\phi^\pi)]$.*

Lemma D.1 proves the unbiasedness of the four OPE methods presented in Section 3.3 when applied to the abstract state space, under the corresponding irrelevance conditions. Notably, DRL requires weaker irrelevance conditions compared to the Q-function-based method and MIS, owing to its inherent double robustness property.

## D.1 PROOF OF LEMMA D.1

We prove Lemma D.1 in this subsection. We first prove that under $Q^\pi$-, $\rho^\pi$- or $w^\pi$-irrelevance, the corresponding methods remain unbiased when applied to the abstract state space:

- **Unbiasedness under $Q^\pi$-irrelevance**. By definition, $Q^\pi$ is the expected return given an initial state $S_1$ and $A_1$. Under $Q^\pi$-irrelevance, the Q-function depends on $S_1$ only through $\phi(S_1)$. It follows that $Q^\pi$ equals the expected return given $\phi(S_1)$ and $A_1$, the latter being $Q_\phi^\pi$ – the Q-function when restricted to the abstract state space, i.e., $Q_\phi^\pi(a, \phi(s)) = \sum_{t\geq 1} \gamma^{t-1}\mathbb{E}^\pi[R_t|A_1 = a, \phi(S_1) = \phi(s)]$. It follows that

$$\mathbb{E}[f_1(Q^\pi)] = \sum_{a,s} \pi(a|s)Q^\pi(a,s)\mathbb{P}(S_1 = s)$$

$$= \sum_{a,s} \pi(a|s)Q_\phi^\pi(a,\phi(s))\mathbb{P}(S_1 = s)$$

$$= \mathbb{E}[f_1(Q_\phi^\pi)].$$

- **Unbiasedness under $\rho^\pi$-irrelevance**. We first establish the equivalence between $\rho^\pi$ and $\rho_\phi^\pi$ – the IS ratio defined on the abstract state space. Under $\rho^\pi$-irrelevance, $\rho^\pi(a, s)$ becomes a constant function of $x = \phi(s)$. Consequently, for any conditional probability mass function (pmf) $f(\bullet|x)$ such that $\sum_{s'\in\phi^{-1}(x)} f(s'|x) = 1$, we have $\rho^\pi(a,s) = \sum_{s'\in\phi^{-1}(x)} f(s'|x)\rho^\pi(a,s')$. By setting $f(\bullet|x)$ to the pmf of $S_t$ given $A_t = a$ and $\phi(S) = x$, it follows that

$$\rho^\pi(a,s) = \sum_{s'\in\phi^{-1}(x)} \mathbb{P}(S_t = s'|A_t = a, \phi(S_t) = x)\rho^\pi(a,s'). \tag{D.1}$$

Notice that

$$\mathbb{P}(S_t = s'|A_t = a, \phi(S_t) = x) = \frac{\mathbb{P}(A_t = a, S_t = s'|\phi(S_t) = x)}{\mathbb{P}(A_t = a|\phi(S_t) = x)}.$$

The denominator equals $b_{\phi,t}(a|x)$, the behavior policy when restricted to the abstract state space at time $t$. Notice that this behavior policy can be non-stationary over time, despite that $b$ being time-invariant. As for the numerator, it is straightforward to show that it equals $b(a|s')\mathbb{P}(S_t = s'|\phi(S_t) = x)$. This together with equation D.1 yields

$$\rho^\pi(a,s) = \sum_{s'\in\phi^{-1}(x)} \frac{\pi(a|s')}{b_{\phi,t}(a|x)}\mathbb{P}(S_t = s'|\phi(S_t) = x) = \frac{\pi_{\phi,t}(a|x)}{b_{\phi,t}(a|x)}, \tag{D.2}$$

where $\pi_{\phi,t}$ denotes the target policy confined on the abstract state space at time $t$, namely, $\pi_{\phi,t}(a|x) = \sum_{s' \in \phi^{-1}(x)} \pi(a|s') \mathbb{P}(S_t = s'|\phi(S_t) = x)$. The last term in equation D.2 is given by $\rho_{\phi,t}^{\pi}$. Consequently, the sequential IS ratio $\rho_{1:t}^{\pi}$ is equal to $\prod_{k=1}^{t} \rho_{\phi,k}^{\pi}(A_k, \phi(S_k))$. This in turn yields $\mathbb{E}[f_2(\rho^{\pi})] = \mathbb{E}[f_2(\rho_{\phi}^{\pi})]$.

- **Unbiasedness under $w^{\pi}$-irrelevance**. Similar to the proof under $\rho^{\pi}$-irrelevance, the key lies in establishing the equivalence between $w^{\pi}(a, s)$ and $w_{\phi}^{\pi}(a, \phi(s))$, the latter being the MIS ratio defined on the abstract state space. Once this has been proven, it is immediate to see that $\mathbb{E}[f_3(w^{\pi})] = \mathbb{E}[f_3(w_{\phi}^{\pi})]$, so that the MIS remains Fisher consistent when applied to the abstract state space.

  As discussed in Section 3.3, to guarantee the unbiasedness of the MIS estimator, we additionally require a stationarity assumption. Under this requirement, for a given state-action pair $(S, A)$ in the offline data, its joint pmf function can be represented as $p_{\infty} \times b$ where $p_{\infty}$ denotes the marginal state distribution under the behavior policy. Additionally, let $p_t^{\pi}$ denote the pmf of $S_t$ generated under the target policy $\pi$. The MIS ratio can be represented by

$$w^{\pi}(a, s) = \frac{(1 - \gamma) \sum_{t \geq 1} \gamma^{t-1} p_t^{\pi}(s) \pi(a|s)}{p_{\infty}(s) b(a|s)}.$$

Similar to equation D.2, under $w^{\pi}$-irreleavance, it follows that

$$
\begin{aligned}
w^{\pi}(a, s) &= (1 - \gamma) \sum_{s' \in \phi^{-1}(x)} \frac{\sum_{t \geq 1} \gamma^{t-1} p_t^{\pi}(s') \pi(a|s')}{p_{\infty}(s') b_{\phi}(a|x)} \mathbb{P}(S = s'|\phi(S) = x) \\
&= \frac{(1 - \gamma) \sum_{s' \in \phi^{-1}(x)} \sum_{t \geq 1} \gamma^{t-1} p_t^{\pi}(s') \pi(a|s')}{p_{\infty}(x) b_{\phi}(a|x)}.
\end{aligned}
$$

Here, the subscript $t$ in $b_{\phi}$ and $S$ is dropped due to stationarity. Additionally, $p_{\infty}(x)$ is used to denote the probability mass function (pmf) of $\phi(S)$, albeit with a slight abuse of notation. Moreover, the numerator represents the discounted visitation probability of $(A, \phi(S))$ under $\pi$. This proves that $w^{\pi}(a, s) = w_{\phi}^{\pi}(a, \phi(s))$.

Finally, we establish the unbiasedness of DRL. According to the doubly robustness property, DRL is Fisher consistent when either $Q^{\pi}$ or $w^{\pi}$ is correctly specified. Under $Q^{\pi}$-irrelevance, we have $Q^{\pi}(a, s) = Q_{\phi}^{\pi}(a, \phi(s))$ and thus DRL remains Fisher consistent when applied to the abstract state space. Similarly, we have $w^{\pi}(a, s) = w_{\phi}^{\pi}(a, \phi(s))$ under $w^{\pi}$-irrelevance, which in turn implies DRL's unbiasedness. This completes the proof.

### D.2 PROOF OF LEMMA 1

We prove Lemma 1 in this subsection. We first show $Q^{\pi}$-irrelevance under model-irrelevance & $\pi$-irrelevance, and prove the Fisher consistency of the Q-function-based method. Then, we establish Fisher consistency of MIS. Next, we derive Fisher consistency of SIS. Finally, we prove the Fisher consistency of DRL.

- **Fisher consistency of Q-function-based method**. We first use the induction method to prove that

$$Q^{\pi}(a, s^{(1)}) = Q^{\pi}(a, s^{(2)}), \tag{D.3}$$

whenever $s^{(1)}$ and $s^{(2)}$ satisfy $\phi(s^{(1)}) = \phi(s^{(2)})$. This demonstrates $Q^{\pi}$-irrelevance, which further implies $\mathbb{E}[f_1(Q^{\pi})] = \mathbb{E}[f_1(Q_{\phi}^{\pi})]$ according to Lemma D.1. We next establish the identifiability of $Q_{\phi}^{\pi}$.

Define

$$Q_j^{\pi}(a, s) = \mathbb{E}^{\pi} \left[ \sum_{t=1}^{j} \gamma^{t-1} R_t | S_1 = s, A_1 = a \right], \text{ and}$$

$$V_j^{\pi}(s) = \mathbb{E}^{\pi} \left[ \sum_{t=1}^{j} \gamma^{t-1} R_t | S_1 = s \right].$$

Under reward-irrelevance, we have

$$Q_1^\pi(a, s^{(1)}) = \mathbb{E}^\pi \left[ R_1 | S_1 = s^{(1)}, A_1 = a \right]$$
$$= \mathcal{R}(a, s^{(1)})$$
$$= \mathcal{R}(a, s^{(2)})$$
$$= Q_1^\pi(a, s^{(2)}).$$

Together with $\pi$-irrelevance, we obtain that

$$V_1^\pi(s^{(1)}) = \sum_{a \in \mathcal{A}} Q_1^\pi(a, s^{(1)}) \pi(a | s^{(1)})$$
$$= \sum_{a \in \mathcal{A}} Q_1^\pi(a, s^{(2)}) \pi(a | s^{(2)})$$
$$= V_1^\pi(s^{(2)}).$$

Suppose we have shown that the following holds for any $j < T$,

$$Q_j^\pi(a, s^{(1)}) = Q_j^\pi(a, s^{(2)}) \text{ and } V_j^\pi(s^{(1)}) = V_j^\pi(s^{(2)}), \tag{D.4}$$

whenever $\phi(s^{(1)}) = \phi(s^{(2)})$. Our goal is to show equation D.4 holds with $j = T$.
We similarly define $Q_{j,\phi}^\pi$ and $V_{j,\phi}^\pi$ as the Q- and value functions on the abstract state space. Similar to the proof of Theorem D.1, we can show that $Q_j^\pi = Q_{j,\phi}^\pi$ and $V_j^\pi = V_{j,\phi}^\pi$ for any $j < T$.
Direct calculations yield

$$Q_T^\pi(a, s^{(1)}) = \mathbb{E}^\pi \left[ \sum_{t=1}^{T} \gamma^{t-1} R_t | S_1 = s^{(1)}, A_1 = a \right]$$

$$= \mathbb{E}^\pi \left[ \sum_{t=2}^{T} \gamma^{t-1} R_t | S_1 = s^{(1)}, A_1 = a \right] + \mathcal{R}(a, s^{(1)})$$

$$= \mathbb{E}^\pi \sum_{s' \in \mathcal{S}} \left[ \sum_{t=2}^{T} \gamma^{t-1} R_t | S_2 = s' \right] \mathcal{T}(s' | s^{(1)}, a) + \mathcal{R}(a, s^{(1)})$$

$$= \gamma \mathbb{E}^\pi \sum_{x' \in \mathcal{X}} \sum_{s' \in \phi^{-1}(x')} \left[ \sum_{t=2}^{T} \gamma^{t-2} R_t | S_2 = s' \right] \mathcal{T}(s' | s^{(1)}, a) + \mathcal{R}(a, s^{(1)})$$

$$= \gamma \sum_{x' \in \mathcal{X}} \sum_{s' \in \phi^{-1}(x')} V_{T-1}^\pi(s') \mathcal{T}(s' | s^{(1)}, a) + \mathcal{R}(a, s^{(1)})$$

$$= \gamma \sum_{x' \in \mathcal{X}} \sum_{s' \in \phi^{-1}(x')} V_{T-1}^\pi(s') \mathcal{T}(s' | s^{(2)}, a) + \mathcal{R}(a, s^{(2)})$$

$$= Q_T^\pi(a, s^{(2)}),$$

where the second last equation holds due to transition-irrelevance in equation 3 and equation D.4, which states that $V_{T-1}^\pi(s')$ is constant as a function of $s' \in \phi^{-1}(x')$.
This together with $\pi$-irrelevance proves $V_T^\pi$-irrelevance. By induction, we have shown that equation D.4 holds for any $j \geq 1$. Under the boundness assumption in Assumption 1, $Q_j^\pi \to Q^\pi$ as $j \to \infty$. We thus obtain equation D.3, which yields $Q^\pi$-irrelevance.
Next, we prove the identifiability of $Q_\phi^\pi$. Similarly, we define

$$Q_{j,\phi}^\pi(a, x) = \sum_{t=1}^{j} \gamma^{t-1} \mathbb{E}^\pi \left[ R_t | \phi(S_1) = x, A_1 = a \right]. \tag{D.5}$$

By setting $j = 1$, it reduces to $\mathbb{E}^\pi[R_1 | \phi(S_1) = x, A_1 = a]$. Under the MDP model assumption, the conditional mean of the immediate reward depends solely on the current state-action pair, independent of past history. This together with the reward-irrelevance condition further implies that the conditional mean of the reward depends solely on the abstract-state-action pair. Consequently,

$$\mathbb{E}^\pi[R_1 | \phi(S_1) = x, A_1 = a] = \underbrace{\mathbb{E}[R_1 | \phi(S_1) = x, A_1 = a]}_{\mathcal{R}_\phi(a,x)}.$$

Notice that the expectation on the right-hand-side (RHS) is defined with respect to the behavior policy. It can thus be consistently estimated using the offline data under the coverage assumption in Assumption 2. This yields the identifiability of $Q_{1,\phi}^\pi$.

Similarly, we can show that

$$\mathbb{P}^\pi[\phi(S_2) = x'|A_1 = a, \phi(S_1) = x] = \underbrace{\mathbb{P}[\phi(S_2) = x'|A_1 = a, \phi(S_1) = x]}_{\mathcal{T}_\phi(x'|a,x)},$$

under transition-irrelevance, which establishes the identifiability of the left-hand-side (LHS).

Notice that for any $j \geq 1$, under the MDP model assumption, $Q_{j,\phi}^\pi$ can be represented using $\mathbb{E}^\pi[R_1|\phi(S_1) = x, A_1 = a]$ and $\mathbb{P}^\pi[\phi(S_2) = x'|A_1 = a, \phi(S_1) = x]$. Both have been proven identifiable. This the establishes identifiability of $Q_{j,\phi}^\pi$. Again, by letting $j \to \infty$, we obtain the identifiability of $Q_\phi^\pi$ under the boundedness assumption in Assumption 1. The proof is hence completed.

- **Fisher consistency of MIS**. We use $p_{t,\phi}^\pi(a, x)$ to denote the probability $\mathbb{P}^\pi(A_t = a, \phi(S_t) = x)$ and $p_t^\pi(s)$ to denote $\mathbb{P}^\pi(S_t = s)$. Under the stationary assumption, direct calculations yield

$$\mathbb{E}[f_3(w_\phi^\pi)] = \mathbb{E}\left[(1-\gamma)^{-1} w_\phi^\pi(A, \phi(S))R\right]$$

$$= \mathbb{E}\left[(1-\gamma)^{-1} w_\phi^\pi(A, \phi(S))\mathcal{R}(A, S)\right]$$

$$= \mathbb{E}\left[(1-\gamma)^{-1} w_\phi^\pi(A, \phi(S)) \underbrace{\mathcal{R}_\phi(A, \phi(S))}_{\text{reward-irrelevant}}\right]$$

$$= \sum_{a \in \mathcal{A}, x \in \mathcal{X}} \sum_{t=1}^{+\infty} \gamma^{t-1} p_t^\pi(a, x)\mathcal{R}_\phi(a, x)$$

$$= \sum_{a \in \mathcal{A}, x \in \mathcal{X}} \sum_{s \in \phi^{-1}(x)} \sum_{t=1}^{+\infty} \gamma^{t-1} \pi(a|s) p_t^\pi(s)\mathcal{R}(a, s)$$

$$= \sum_{t=1}^{+\infty} \gamma^{t-1} \mathbb{E}^\pi(R_t)$$

$$= \mathbb{E}[f_3(w^\pi)].$$

To complete the proof, it remains to establish the identifiability of $w_\phi^\pi$.

Under the stationarity assumption in Assumption 3, $\omega_\phi^\pi$ can be represented by

$$\frac{\sum_{t \geq 1} \gamma^{t-1} p_{t,\phi}^\pi(a, x)}{\mathbb{P}(A_1 = a, \phi(S_1) = x)}.$$

It is immediate to see that the denominator is identifiable, as the probability is calculated with respect to the behavior policy. It suffices to show that for any $t \geq 1$, $p_t^\pi$ is identifiable as well. Under transition-irrelevance, the data triplets $(\phi(S), A, R)$ forms an MDP, satisfying the Markov assumption. As such, we can rewrite $p_t^\pi(a_t, x_t)$ as

$$\sum_{\substack{a_1, \cdots, a_{t-1} \in \mathcal{A} \\ x_1, \cdots, x_{t-1} \in \mathcal{X}}} \rho_{0,\phi}(x_1) \prod_{k=1}^{t-1} \left[\pi_\phi(a_k|x_k)\mathcal{T}_\phi(x_{k+1}|a_k, x_k)\right] \pi_\phi(a_t|x_t),$$

where $\rho_{0,\phi}$ denotes the pmf of $\phi(S_1)$ which is independent of $\pi$, and both $\pi_\phi$ and $\mathcal{T}_\phi$ are identifiable under $\pi$- and transition-irrelevance, respectively. This proves the identifiability of $p_t^\pi$, and hence, the identifiability of $w_\phi^\pi$.

- **Fisher consistency of SIS**. Recall that we require the behavior policy to be Markovian, meaning that at any time $t$, $A_t$ is independent of historical observations given $S_t$. A key challenge in state abstraction for the SIS estimator is that, after abstraction, the behavior policy can be history-dependent, leading to the inconsistency of SIS. Toward that end, we employ a history-dependent IS ratio to address this challenge. Specifically, let $\rho_{j,\phi}^\pi$ denote

$$\rho_{j,\phi}^\pi = \frac{\pi_\phi(A_j|\phi(S_j))}{b_{j,\phi}(A_j|\phi(S_j), A_{j-1}, \phi(S_{j-1}), \dots, A_1, \phi(S_1))}. \tag{D.6}$$

Under $\pi$-irrelevance, the numerator is well-defined and identifiable. However, unlike the standard IS ratio where the denominator depends solely on the current state, the denominator in equation D.6 depends on the entire history. Notice that the denominator is essentially the data distribution of $A_j$ given $\phi(S_j), A_{j-1}, \phi(S_{j-1}), \ldots, A_1, \phi(S_1)$, it is thus identifiable from the offline data as well. Under the coverage assumption in Assumption 2, the behavior policy is bounded away from zero. Since the behavior policy is stationary, this conditional pmf can be represented by

$$\mathbb{E}[b(\bullet|S_j)|\phi(S_j), A_{j-1}, \phi(S_{j-1}), \ldots, A_1, \phi(S_1)],$$

which is bounded away from zero as well. Consequently, $\rho_{t,\phi}^\pi$ is bounded and identifiable.

Let $\rho_{1:t,\phi}^\pi$ denote the SIS ratio $\prod_{j=1}^t \rho_{j,\phi}^\pi$. It suffices to show

$$\mathbb{E}(\rho_{1:t}^\pi R_t) = \mathbb{E}(\rho_{1:t,\phi}^\pi R_t), \tag{D.7}$$

for any $t$. Under the Markov assumption, $R_t$ is independent of past state-action pairs given $A_t$ and $S_t$. Consequently, the left-hand-side can be represented as

$$\mathbb{E}[\mathbb{E}(\rho_{1:t-1}^\pi|A_t, S_t)\rho^\pi(A_t, S_t)R_t].$$

Additionally, since the generation $A_t$ depends only on $S_t$, the inner expectation equals $\mathbb{E}(\rho_{1:t-1}^\pi|S_t)$ which can be further shown to equal to $p_t^\pi(S_t)/p_\infty(S_t)$. This allows us to represent the left-hand-side of equation D.7 by

$$\mathbb{E}\Big[\frac{p_t^\pi(S_t)}{p_\infty(S_t)}\rho^\pi(A_t, S_t)R_t\Big]. \tag{D.8}$$

Using similar arguments to the proof of Fisher consistency of MIS estimator, under reward-irrelevance, equation D.8 can be shown to equal to

$$\sum_{\substack{a_1,\cdots,a_t\in\mathcal{A} \\ x_1,\cdots,x_t\in\mathcal{X}}} \rho_0(x_1) \prod_{k=1}^{t-1} \Big[\pi_\phi(a_k|x_k)\mathcal{T}_\phi(x_{k+1}|a_k, x_k)\Big]\pi_\phi(a_t|x_t)\mathcal{R}_\phi(a_t, x_t).$$

Notice that both $\mathcal{T}_\phi$ and $\mathcal{R}_\phi$ independent of the target policy $\pi$. Using the change of measure theorem, we can represent above expression by $\mathbb{E}(\rho_{1:t,\phi}^\pi R_t)$. This completes the proof.

- **Fisher consistency of DRL under model-irrelevance**. Since model-irrelevance and $\pi$-irrelevance imply $Q^\pi$-irrelevance and the identifiability of $Q^\pi$, the conclusion directly follows from the double robustness of DRL and that in the first bullet point.

D.3  PROOF OF THEOREM 1

We establish the Fisher consistencies of SIS, MIS, Q-function-based method and DRL one by one.

- **Fisher consistency of SIS**. Notice that $\rho^\pi$-irrelevance directly follows from the definition of backward-model-irrelevance and $\pi$-irrelevance. It follows from Lemma D.1 that $\mathbb{E}[f_2(\rho^\pi)] = \mathbb{E}[f_2(\rho_\phi^\pi)]$.
  Additionally, under $\pi$-irrelevance and behavior-policy-irrelevance, both the numerator and the denominator of the IS ratio $\rho_\phi^\pi$ are identifiable. Consequently, $\rho_\phi^\pi$ is identifiable as well. This establishes the Fisher consistency of SIS.
- **Fisher consistency of MIS**. We first establish the $w^\pi$-irrelevance. We next establish the identifiability of $w_\phi^\pi$.
  To prove the $w^\pi$-irrelevance, we begin by defining the marginal density ratio at a given time $t$ as

$$w_t^\pi(a, s) = \frac{\mathbb{P}^\pi(A_t = a, S_t = s)}{\mathbb{P}(A_t = a, S_t = s)}.$$

Under the stationarity assumption, the denominator is independent of $t$. Notice that $w^\pi(a, s) = (1-\gamma)\sum_{t=1}^\infty \gamma^{t-1}w_t^\pi(a, s)$. Hence, it is sufficient to prove that $\phi$ is $w_t^\pi$-irrelevance, for any $t$. We prove this by induction. First, when $t = 1$, $w_t^\pi$ is reduced to $\rho^\pi$. Consequently, $w_1^\pi$-irrelevance is readily obtained by backward-model-irrelevance and $\pi$-irrelevance.

Second, suppose we have established $w_t^\pi$-irrelevance. We aim to show $w_{t+1}^\pi$-irrelevance. With some calculations, we obtain that

$$\mathbb{E}[w_t^\pi(A_t, S_t)\rho^\pi(A_{t+1}, S_{t+1})|A_{t+1} = a, S_{t+1} = s]$$

$$= \sum_{a',s'} w_t^\pi(a', s')\frac{\pi(a|s)}{b(a|s)}\mathbb{P}(A_t = a', S_t = s'|A_{t+1} = a, S_{t+1} = s)$$

$$= \sum_{a',s'} \frac{\mathbb{P}^\pi(A_t = a', S_t = s')}{\mathbb{P}(A_t = a', S_t = s')}\frac{\pi(a|s)}{b(a|s)}\frac{\mathbb{P}(A_t = a', S_t = s', A_{t+1} = a, S_{t+1} = s)}{\mathbb{P}(A_{t+1} = a, S_{t+1} = s)}$$

$$= \sum_{a',s'} \mathbb{P}^\pi(A_t = a', S_t = s')\frac{\pi(a|s)}{b(a|s)} \times \frac{\mathbb{P}(A_{t+1} = a|S_{t+1} = s, A_t = a', S_t = s')}{\mathbb{P}(A_{t+1} = a, S_{t+1} = s)}\mathbb{P}(S_{t+1} = s|A_t = a', S_t = s)$$

$$= \sum_{a',s'} \mathbb{P}^\pi(A_t = a', S_t = s')\frac{\pi(a|s)}{\mathbb{P}(A_{t+1} = a, S_{t+1} = s)}\mathbb{P}(S_{t+1} = s|A_t = a', S_t = s)$$

$$= \frac{\mathbb{P}^\pi(A_{t+1} = a, S_{t+1} = s)}{\mathbb{P}(A_{t+1} = a, S_{t+1} = s)}$$

$$= w_{t+1}^\pi(a, s), \tag{D.9}$$

where the third last equality is due to that the behavior policy is stationary. This establishes the link between $w_t^\pi$, $\rho^\pi$ and $w_{t+1}^\pi$.

To prove $w_{t+1}^\pi$-irrelevance, we first prove the following equation holds:

$$\sum_{s'\in\phi^{-1}(x')} \mathbb{P}(A_t = a', S_t = s'|A_{t+1} = a, S_{t+1} = s^{(1)})$$

$$= \sum_{s'\in\phi^{-1}(x')} \mathbb{P}(A_t = a', S_t = s'|A_{t+1} = a, S_{t+1} = s^{(2)}), \tag{D.10}$$

whenever $\phi(s^{(1)}) = \phi(s^{(2)})$.

Indeed, by equation 5, we obtain that

$$\sum_{s'\in\phi^{-1}(x')} \mathbb{P}(A_t = a', S_t = s'|A_{t+1} = a, S_{t+1} = s^{(1)})$$

$$= \sum_{s'\in\phi^{-1}(x')} \frac{\mathbb{P}(A_t = a', S_t = s', A_{t+1} = a, S_{t+1} = s^{(1)})}{\mathbb{P}(A_{t+1} = a, S_{t+1} = s^{(1)})}$$

$$= \sum_{s'\in\phi^{-1}(x')} \frac{\mathbb{P}(A_{t+1} = a|S_{t+1} = s^{(1)}, A_t = a', S_t = s')}{\mathbb{P}(A_{t+1} = a, S_{t+1} = s^{(1)})}\mathbb{P}(S_{t+1} = s^{(1)}, A_t = a', S_t = s')$$

$$= \sum_{s'\in\phi^{-1}(x')} \mathbb{P}(A_t = a', S_t = s'|S_{t+1} = s^{(1)})$$

$$= \sum_{s'\in\phi^{-1}(x')} \mathbb{P}(A_t = a', S_t = s'|S_{t+1} = s^{(2)})$$

$$= \sum_{s'\in\phi^{-1}(x')} \mathbb{P}(A_t = a', S_t = s'|A_{t+1} = a, S_{t+1} = s^{(2)}).$$

Consequently,

$$w_{t+1}^\pi(a, s^{(1)})$$

$$= \sum_{a',s'} w_t^\pi(a', s')\rho^\pi(a, s^{(1)})\mathbb{P}(A_t = a', S_t = s'|A_{t+1} = a, S_{t+1} = s^{(1)})$$

$$= \sum_{a',s'} w_t^\pi(a', s')\rho^\pi(a, s^{(2)})\mathbb{P}(A_t = a', S_t = s'|A_{t+1} = a, S_{t+1} = s^{(2)})$$

$$= w_{t+1}^\pi(a, s^{(2)}),$$

where the second last equation follows from $w_t^\pi$-irrelevance, $\rho^\pi$-irrelevance and equation D.10. This yields $w_{t+1}^\pi$-irrelevance, and subsequently $w^\pi$-irrelevance, by induction. By Lemma D.1, $w^\pi$-irrelevance further yields $\mathbb{E}[f_3(w^\pi)] = \mathbb{E}[f_3(w_\phi^\pi)]$.

It remains to prove the identifiability of $w_\phi^\pi$. We similarly define

$$w_{t,\phi}^\pi(a, x) = \frac{\mathbb{P}^\pi(A_t = a, \phi(S_t) = x)}{\mathbb{P}(A_t = a, \phi(S_t) = x)}.$$

It follows that $w_\phi^\pi = \sum_{t \geq 1} \gamma^{t-1} w_{t,\phi}^\pi$. Again, $w_{1,\phi}^\pi$ corresponds to $\rho_\phi^\pi$, which is identifiable under backward-model-irrelevance and $\pi$-irrelevance.

Based on the aforementioned arguments, we can show that

$$w_{t+1,\phi}^\pi(a, x) = \sum_{a', x'} w_{t,\phi}^\pi(a', x') \rho_\phi(a|x) \mathbb{P}(A_t = a', \phi(S_t) = x' | A_{t+1} = a, \phi(S_{t+1}) = x),$$

where the last term on the RHS is well-defined according to equation D.10. Suppose we have shown the identifiability of $w_{t,\phi}^\pi$. Then each term on the RHS is identifiable. This proves the identifiability of $w_{t+1,\phi}^\pi$. By induction, $w_{t,\phi}^\pi$ is identifiable for each $t \geq 1$.

According to the coverage and stationarity assumptions in Assumptions 2 and 3, the denominators in $\{w_{t,\phi}^\pi\}$ are bounded away from zero. Consequently, $\{w_{t,\phi}^\pi\}$ are uniformly bounded. By letting $t \to \infty$, we obtain the identifiability of $w_\phi^\pi$. The proof is thus completed.

- **Fisher consistency of Q-function-based method**. We first show that $\mathbb{E}[f_1(Q_\phi^\pi)] = \mathbb{E}[f_1(Q^\pi)]$ under $\pi$-irrelevance. This is immediate by noting that

$$\mathbb{E}[f_1(Q^\pi)] = J(\pi) = \sum_{t \geq 1} \gamma^{t-1} \mathbb{E}^\pi(R_t) = \sum_{a,x} \sum_{t \geq 1} \gamma^{t-1} \mathbb{E}^\pi(R_t | A_1 = a, \phi(S_1) = x)$$

$$\times \mathbb{P}^\pi(A_1 = a | \phi(S_1) = x) \mathbb{P}(\phi(S_1) = x) = \mathbb{E}[f_1(Q_\phi^\pi)],$$

where the first term on the second line equals $\pi(a|s)$ for any $s$ such that $\phi(s) = x$, under $\pi$-irrelevance.

It remains to prove the identifiability of $Q_\phi^\pi$ under $\pi$- and backward-model-irrelevance. First, we establish the identifiability of $\mathbb{E}^\pi(R_1 | A_1 = a, \phi(S_1) = x)$. By definition

$$\mathbb{E}^\pi(R_1 | A_1 = a, \phi(S_1) = x) = \frac{\mathbb{E}^\pi[R_1 \mathbb{I}(A_1 = a, \phi(S_1) = x)]}{\mathbb{P}^\pi(A_1 = a, \phi(S_1) = x)}.$$

Using the change of measure theorem, the numerator equals

$$\mathbb{E}\Big[\rho^\pi(a|S_1) R_1 \mathbb{I}(A_1 = a, \phi(S_1) = x)\Big] = \mathbb{E}\Big[\rho_\phi^\pi(a|x) R_1 \mathbb{I}(A_1 = a, \phi(S_1) = x)\Big]$$

$$= \rho_\phi^\pi(a|x) \mathbb{E}\Big[R_1 \mathbb{I}(A_1 = a, \phi(S_1) = x)\Big],$$

where the first equation holds due to $\pi$- and behavior-policy-irrelevance. Notice that the denominator equals $\mathbb{P}(\phi(S_1) = x) \pi_\phi(a|x)$, it follows that

$$\mathbb{E}^\pi(R_1 | A_1 = a, \phi(S_1) = x) = \mathbb{E}(R_1 | A_1 = a, \phi(S_1) = x),$$

which is identiable from the data.

Similarly, one can show that $\mathbb{P}^\pi(\phi(S_2) = x' | A_1 = a, \phi(S_1) = x) = \mathbb{P}(\phi(S_2) = x' | A_1 = a, \phi(S_1) = x)$ is identifiable as well.

Now, the identifiability can be readily obtained if we show $(\phi(S_t), A_t, R_t)_{t \geq 1}$ remains an MDP. In that case, standard Q-learning algorithms can be applied to such a reduced MDP to consistently identify $Q_\phi^\pi$. Such an MDP property will be proven in Section D.4.2 under a more challenging setting that allows the behavior policy to be history-dependent.

- **Fisher consistency of DRL**. Due to the double robustness property of DRL, the conclusion directly follows from the last conclusion of Theorem D.1 and the first two conclusions of Theorem 1.

## D.4 PROOF OF THEOREM 2

First, we notice that according to the DRL's double robustness property, its Fisher consistency is achieved when either the MIS or the Q-function-based estimator is Fisher consistent. Consequently, it suffices to prove the Fisher consistencies of the rest three estimators.

Additionally, at the first iteration, these Fisher consistencies directly follows from Lemma 1. Consequently, it suffices to prove the Fisher consistencies at later iterations. Below, we first prove the Fisher consistencies of SIS, MIS and Q-function-based estimator at the second iteration. Next, we prove the resulting abstraction is a Markov state abstraction (Allen et al., 2021) in that the data generating process when confined to the abstract state space remains an MDP. This together with Lemma 1 proves the Fisher consistencies at the third iteration. Using similar arguments, we can establish the Fisher consistencies at any $K > 3$ iterations. The proof can thus be completed.

### D.4.1 FISHER CONSISTENCIES AT THE 2ND ITERATION

It is worthwhile mentioning that at the second iteration, the refined backward-model-irrelevance condition is defined with respect to the abstract state space induced by the forward abstraction $\phi_1$ at the first iteration instead of the ground state space. In particular, we require

$$b_{\phi_1,t}(a_t|x_t^{(1)}, a_{t-1}, x_{t-1}^{(1)}, \cdots, x_1^{(1)}) = b_{\phi_1,t}(a_t|x_t^{(2)}, a_{t-1}, x_{t-1}^{(2)}, \cdots, x_1^{(2)}), \quad \text{(D.11)}$$

for any $t$ and $\{a_t\}_t$, whenever $\phi_2(x_t^{(1)}) = \phi_2(x_t^{(1)})$ for any $\{x_t^{(1)}\}_t$ and $\{x_t^{(2)}\}_t$, where $b_{\phi_1,t}$ denote the history-dependent behavior policy (see also the denominator of Equation D.6), and

$$\sum_{x \in \phi_2^{-1}(x_2)} \mathbb{P}(A_t = a, \phi_1(S_t) = x|\phi_1(S_{t+1}) = x^{(1)})$$
$$= \sum_{x \in \phi_2^{-1}(x_2)} \mathbb{P}(A_t = a, \phi_1(S_t) = x|\phi_1(S_{t+1}) = x^{(2)}), \quad \text{(D.12)}$$

whenever $\phi_2(x^{(1)}) = \phi_2(x^{(2)})$.

In the following, we prove the Fisher consistencies of SIS, MIS and Q-function-based method one by one:

- **Fisher consistency of SIS**. When restricting to the abstract state space induced by $\phi_1$, the resulting behavior policy is not guaranteed to be Markovian. To address this challenge, SIS employs the history-dependent IS ratio defined in Equation D.6 to maintain consistency. Let $\rho_{1:t,\phi_1}^\pi$ and $\rho_{1:t,\phi_2\circ\phi_1}^\pi$ denote the history-dependent SIS ratios at the first and second iterations, respectively. Under $\pi$-irrelevance and the refined history-dependent-behavior-policy-irrelevance (see equation D.11), it is immediate to see that $\rho_{1:t,\phi_1}^\pi = \rho_{1:t,\phi_2\circ\phi_1}^\pi$ so that $\rho^\pi$-irrelevance is achieved at the second iteration. This in turn validates the unbiasedness of the SIS estimator based on $\{\rho_{1:t,\phi_2\circ\phi_1}^\pi\}_t$.
  Finally, notice that the denominators in $\rho_{1:t,\phi_2\circ\phi_1}^\pi$ are identifiable since these probabilities are defined with respect to the offline data distribution. Meanwhile, under $\pi$-irrelevance, the numerator is identifiable as well. This proves the identifiability of these history-dependent IS ratios. The Fisher consistency of SIS thus follows.
- **Fisher consistency of MIS**. We first show that the abstraction produced by DSA at the second iteration achieves $w^\pi$-irrelevance, i.e., $w_{\phi_1}^\pi = w_{\phi_2\circ\phi_1}^\pi$. We next establish the identifiability of the MIS ratio $w_{\phi_2\circ\phi_1}^\pi$.
  The proof is very similar to that of Theorem 1. Specifically, define

$$w_{t,\phi_1}^\pi(a, x) = \frac{\mathbb{P}^\pi(A_t = a, \phi(S_t) = x)}{\mathbb{P}(A_t = a, \phi(S_t) = x)},$$

we have $w_{\phi_1}^\pi = (1 - \gamma) \sum_{t \geq 1} \gamma^{t-1} w_{t,\phi_1}^\pi$. It suffices to establish the irrelevance in $w_{t,\phi_1}^\pi$ for any $t$. When $t = 1$, $w_{1,\phi_1}^\pi$ is reduced to the IS ratio $\pi_{\phi_1}(a|x)/b_{1,\phi_1}(a, x)$. Under $\pi$-irrelevance and behavior-policy-irrelevance (by setting $j$ in Equation D.11 to 1), the numerator $\pi_{\phi_1}$ and denominator $b_{1,\phi_1}$ equal $\pi_{\phi_2\circ\phi_1}$ and $b_{1,\phi_2\circ\phi_1}$ (the behavior policy when restricting to the abstract state space produced by DSA at the 2nd iteration), respectively. This establishes the irrelevance in $w_{1,\phi_1}^\pi$. Suppose we have proven the irrelevance in $w_{t,\phi_1}^\pi$, we aim to show the irrelevance in $w_{t+1,\phi_1}^\pi$. Under the stationarity assumption in Assumption 3, by setting $j$ in Equation 6 to 2, we obtain that

$$\mathbb{P}(A_t = a_2|\phi_1(S_t) = x_2^{(1)}, A_{t-1} = a_1, \phi_1(S_{t-1}) = x_1^{(1)})$$
$$= \mathbb{P}(A_t = a_2|\phi_1(S_t) = x_2^{(2)}, A_{t-1} = a_1, \phi_1(S_{t-1}) = x_1^{(2)}), \quad \text{(D.13)}$$

for any $t$, $a_1$ and $a_2$, whenever $\phi_2(x_1^{(1)}) = \phi_2(x_1^{(2)})$ and $\phi_2(x_2^{(1)}) = \phi_2(x_2^{(2)})$.

Let $X_t$ denote $\phi_1(S_t)$ for any $t$. We next claim that

$$w_{t+1,\phi_1}(a, x) = \mathbb{E}\left[w_{t,\phi_1}(A_t, X_t) \frac{\pi_{\phi_1}(A_t|X_t)}{b_{2,\phi_1}(A_{t+1}|X_{t+1}, A_t, X_t)} \Big| A_{t+1} = a, X_{t+1} = x\right]. \quad \text{(D.14)}$$

Notice that this formula is very similar to equation D.9. The only difference lies in that the denominator of the IS ratio on the RHS is no longer Markovian. Rather, it depends on the current state as well as the previous state-action pair. Meanwhile, equation D.14 can be proven using similar arguments to equation D.9.

Based on equation D.14, we are ready to establish the irrelevance in $w_{t+1,\phi}^\pi$. In particular, looking at the RHS of equation D.14, both $w_{t,\phi_1}$ and the IS ratio $\pi_{\phi_1}/b_{2,\phi_1}$ depend on $X_t$ and $X_{t+1}$ only through their abstractions $\phi_2(X_t)$ and $\phi_2(X_{t+1})$. Meanwhile, the conditional distribution of $A_t, X_t$ given $A_{t+1}, X_{t+1}$ depends on $X_t$ and $X_{t+1}$ through their abstractions, as well, given equation D.13 and equation D.12. This establishes the irrelevance in $w_{t+1,\phi_1}$. By induction, we have proven the irrelevance in $w_{t,\phi_1}$ for any $t$. Under the coverage assumption in Assumption 2, these ratios are uniformly bounded. It follows that the limit $\lim_T \sum_{t=1}^T \gamma^{t-1} w_{t,\phi_1}^\pi$ is well-defined. By setting $T \to \infty$, we obtain the irrelevance in $w_{\phi_1}^\pi$.

So far, we have established the $w^\pi$-irrelevance. This in turn yields $\mathbb{E}[f_3(w_{\phi_1}^\pi)] = \mathbb{E}[f_3(w_{\phi_2 \circ \phi_1}^\pi)]$, according to Lemma D.1. It remains to prove the identifiability of $w_{\phi_2 \circ \phi_1}^\pi$. However, this can be proven using similar arguments to the proof of Theorem 1. Specifically, we first observe that $w_{\phi_2 \circ \phi_1}^\pi = \lim_T \sum_{t=1}^T \gamma^{t-1} w_{t,\phi_2 \circ \phi_1}^\pi$. Next, when setting $t = 1$, the identifiability of $w_{1,\phi_2 \circ \phi_1}^\pi$ is readily available, given that of $\rho_{\phi_2 \circ \phi_1}^\pi$. Finally, since $w_{t+1,\phi_2 \circ \phi_1}(a, x_2)$ equals

$$\mathbb{E}\left[w_{t,\phi_2 \circ \phi_1}(A_t, \phi_2(X_t)) \frac{\pi_{\phi_2 \circ \phi_1}(A_t|\phi_2(X_t))}{b_{2,\phi_2 \circ \phi_1}(A_{t+1}|\phi_2(X_{t+1}), A_t, \phi_2(X_t))} \Big| A_{t+1} = a, \phi_2(X_{t+1}) = x_2\right],$$

we can employ similar arguments to the proof of Theorem 1 to prove the identifiability of the above expression, assuming $w_{t,\phi_2 \circ \phi_1}$ is identifiable. By induction, this establishes the identifiability of $w_{\phi_2 \circ \phi_1}$.

- **Fisher consistency of Q-function-based method**. The Fisher consistency of Q-function-based method can be established in a similar manner to that in Theorem 1. Specifically, under $\pi$-irrelevance, it is trivial to show $\mathbb{E}[f_1(Q_{\phi_1}^\pi)] = J(\pi) = \mathbb{E}[f_1(Q_{\phi_2 \circ \phi_1}^\pi)]$. Meanwhile, its identifiability is readily obtained based on the results in the following section, which proves that the process $(\phi_2(\phi_1(S_t)), A_t, R_t)_{t \geq 1}$ remains an MDP.

### D.4.2 BACKWARD ABSTRACTION IS A MARKOV STATE ABSTRACTION

It is equivalent to prove that, when the backward abstraction $\phi$ is obtained by applying the refined backward-model-irrelevance condition to the original MDP $(S_t, A_t, R_t)_{t \geq 1}$ with a history-dependent behavior policy, the reduced process $(\phi(S_t), A_t, R_t)_{t \geq 1}$ remains an MDP.

We start by presenting the following lemma and its proof.

**Lemma D.2** *For any $a, x, t$ and $s_{t+1}, x_{t+1}$ such that $\phi(s_{t+1}) = x_{t+1}$, we have*

$$\sum_{s \in \phi^{-1}(x)} \mathbb{P}(A_t = a, S_t = s|S_{t+1} = s_{t+1}) = \mathbb{P}(A_t = a, \phi(S_t) = x|\phi(S_{t+1}) = x_{t+1}). \quad \text{(D.15)}$$

*Additionally, for any $a_t, s_t, x_t$ such that $\phi(s_{t+1}) = x_{t+1}$, we have*

$$\frac{\mathbb{P}(A_t = a_t|S_t = s_t)}{\mathbb{P}(A_t = a_t|\phi(S_t) = x_t)} = \frac{\mathbb{P}(A_t = a_t|S_t = s_t, \{A_{t-k} = a_{t-k}, \phi(S_{t-k}) = x_{t-k}\}_{k \in G})}{\mathbb{P}(A_t = a_t|\phi(S_t) = x_t, \{A_{t-k} = a_{t-k}, \phi(S_{t-k}) = x_{t-k}\}_{k \in G})}, \quad \text{(D.16)}$$

*for any $G = \{1, 2, \ldots, \ell\}$ with any $\ell \in \{1, 2, \ldots, t-1\}$.*

**Proof of Lemma D.2**. Using similar arguments to equation D.2, we have

$$\mathbb{P}(A_t = a, \phi(S_t) = x | \phi(S_{t+1}) = x_{t+1})$$

$$= \frac{\mathbb{P}(A_t = a, \phi(S_t) = x, \phi(S_{t+1}) = x_{t+1})}{\mathbb{P}(\phi(S_{t+1}) = x_{t+1})}$$

$$= \sum_{s_{t+1} \in \phi^{-1}(x_{t+1})} \frac{\mathbb{P}(A_t = a, \phi(S_t) = x, S_{t+1} = s_{t+1})}{\mathbb{P}(\phi(S_{t+1}) = x_{t+1})}$$

$$= \sum_{s'_{t+1} \in \phi^{-1}(x_{t+1})} \mathbb{P}(A_t = a, \phi(S_t) = x | S_{t+1} = s_{t+1}) \mathbb{P}(S_{t+1} = s'_{t+1} | \phi(S_{t+1}) = x_{t+1})$$

$$= \mathbb{P}(A_t = a, \phi(S_t) = x | S_{t+1} = s_{t+1})$$

$$= \sum_{s \in \phi^{-1}(x)} \mathbb{P}(A_t = a, S_t = s | S_{t+1} = s_{t+1}), \tag{D.17}$$

where the third equation is due to the backward-transition-irrelevance condition, under which $\mathbb{P}(A_t = a, \phi(S_t) = x | S_{t+1} = s_{t+1})$ equals $\mathbb{P}(A_t = a, \phi(S_t) = x | S_{t+1} = s'_{t+1})$. This proves equation D.15.

Next, under the stationarity assumption in Assumption 3 and the history-dependent-behavior-policy-irrelevance condition, we have for any $\{s_\ell^{(1)}\}_\ell$, $\{s_\ell^{(2)}\}_t$ such that $\phi(s_\ell^{(1)}) = \phi(s_\ell^{(2)}) = x_\ell$ for all $\ell \geq 1$ that

$$\mathbb{P}(A_t = a_t | S_t = s_t^{(1)}, \{A_{t-k} = a_{t-k}, S_{t-k} = s_{t-k}^{(1)}\}_{k \in G}))$$

$$= \mathbb{P}(A_t = a_t | S_t = s_t^{(2)}, \{A_{t-k} = a_{t-k}, S_{t-k} = s_{t-k}^{(2)}\}_{k \in G}), \tag{D.18}$$

for any $t$, $\{a_\ell\}_\ell$ and $G$. This in turn yields,

$$\frac{\mathbb{P}(A_t = a_t | S_t = s_t^{(1)})}{\mathbb{P}(A_t = a_t | S_t = s_t^{(2)})}$$

$$= \frac{\mathbb{P}(A_t = a_t | S_t = s_t^{(1)}, \{A_{t-k} = a_{t-k}, S_{t-k} = s_{t-k}^{(1)}\}_G))}{\mathbb{P}(A_t = a_t | S_t = s_t^{(2)}, \{A_{t-k} = a_{t-k}, S_{t-k} = s_{t-k}^{(2)}\}_G)}. \tag{D.19}$$

With some calculations, we have

$$\mathbb{P}(A_t = a_t | S_t = s_t, \{A_{t-k} = a_{t-k}, \phi(S_{t-k}) = x_{t-k}\}_{k \in G})$$

$$= \frac{\mathbb{P}(A_t = a_t, \{A_{t-k} = a_{t-k}, \phi(S_{t-k}) = x_{t-k}\}_{k \in G} | S_t = s_t)}{\mathbb{P}(\{A_{t-k} = a_{t-k}, \phi(S_{t-k}) = x_{t-k}\}_{k \in G} | S_t = s_t)}$$

$$= \sum_{s_{t-k} \in \phi^{-1}(x_{t-k}), k \in G} \frac{\mathbb{P}(A_t = a_t, \{A_{t-k} = a_{t-k}, S_{t-k} = s_{t-k}\}_{k \in G} | S_t = s_t)}{\mathbb{P}(\{A_{t-k} = a_{t-k}, \phi(S_{t-k}) = x_{t-k}\}_{k \in G} | S_t = s_t)}$$

$$= \sum_{s_{t-k} \in \phi^{-1}(x_{t-k}), k \in G} \frac{\mathbb{P}(\{A_{t-k} = a_{t-k}, S_{t-k} = s_{t-k}\}_{k \in G} | S_t = s_t)}{\mathbb{P}(\{A_{t-k} = a_{t-k}, \phi(S_{t-k}) = x_{t-k}\}_{k \in G} | S_t = s_t)}$$

$$\times \mathbb{P}(A_t = a_t | S_t = s_t, \{A_{t-k} = a_{t-k}, S_{t-k} = s_{t-k}^{(1)}\}_{k \in G})$$

$$= \mathbb{P}(A_t = a_t | S_t = s_t, \{A_{t-k} = a_{t-k}, S_{t-k} = s_{t-k}^{(1)}\}_{k \in G}), \tag{D.20}$$

where the last equation follows from equation D.18.

Combing equation D.19 with equation D.20, we obtain that

$$\frac{\mathbb{P}(A_t = a_t | S_t = s_t^{(1)}, \{A_{t-k} = a_{t-k}, \phi(S_{t-k}) = x_{t-k}\}_{k \in G})}{\mathbb{P}(A_t = a_t | S_t = s_t^{(2)}, \{A_{t-k} = a_{t-k}, \phi(S_{t-k}) = x_{t-k}\}_{k \in G})}$$

$$= \frac{\mathbb{P}(A_t = a_t | S_t = s_t^{(1)}, \{A_{t-k} = a_{t-k}, S_{t-k} = s_{t-k}^{(1)}\}_{k \in G})}{\mathbb{P}(A_t = a_t | S_t = s_t^{(2)}, \{A_{t-k} = a_{t-k}, S_{t-k} = s_{t-k}^{(2)}\}_{k \in G})}$$

$$= \frac{\mathbb{P}(A_t = a_t | S_t = s_t^{(1)})}{\mathbb{P}(A_t = a_t | S_t = s_t^{(2)})},$$

or equivalently,

$$\frac{\mathbb{P}(A_t = a_t | S_t = s_t^{(1)})}{\mathbb{P}(A_t = a_t | S_t = s_t^{(1)}, \{A_{t-k} = a_{t-k}, \phi(S_{t-k}) = x_{t-k}\}_{k \in G})} \tag{D.21}$$
$$= \frac{\mathbb{P}(A_t = a_t | S_t = s_t^{(2)})}{\mathbb{P}(A_t = a_t | S_t = s_t^{(2)}, \{A_{t-k} = a_{t-k}, \phi(S_{t-k}) = x_{t-k}\}_{k \in G})}.$$

Using similar arguments to equation D.2 and equation D.17, the LHS can be represented by

$$\frac{\mathbb{P}(A_t = a_t | \phi(S_t) = x_t)}{\mathbb{P}(A_t = a_t | \phi(S_t) = x_t, \{A_{t-k} = a_{t-k}, \phi(S_{t-k}) = x_{t-k}\}_{k \in G})}$$

equation D.16 follows directly from equation D.21.

**Proof of the Markov property**. We next prove that the refined backward abstraction is indeed an MSA, despite that the behavior policy is no longer Markovian. Toward that end, we first show that the evolution of $\phi(S_t)$ remains Markovian. Specifically, we aim to show

$$(A_{t-k}, \phi(S_{t-k}))_{1 \le k \le t-1} \perp\!\!\!\perp S_{t+1} | (\phi(S_t), A_t). \tag{D.22}$$

Indeed, by setting the time index $t$ in equation D.15 to $t + 1$, we obtain that

$$\frac{\mathbb{P}(S_t = s_t | A_{t-1} = a_{t-1}, \phi(S_{t-1}) = x_{t-1})}{\mathbb{P}(\phi(S_t) = x_t | A_{t-1} = a_{t-1}, \phi(S_{t-1}) = x_{t-1})} = \frac{\mathbb{P}(S_t = s_t)}{\mathbb{P}(\phi(S_t) = x_t)}. \tag{D.23}$$

Combing equation D.23 with equation D.16, we have

$$\frac{\mathbb{P}(S_t = s_t | A_{t-1} = a_{t-1}, \phi(S_{t-1}) = x_{t-1}) \mathbb{P}(A_t = a_t | S_t = s_t, A_{t-1} = a_{t-1}, \phi(S_{t-1}) = x_{t-1})}{\mathbb{P}(\phi(S_t) = x_t | A_{t-1} = a_{t-1}, \phi(S_{t-1}) = x_{t-1}) \mathbb{P}(A_t = a_t | \phi(S_t) = x_t, A_{t-1} = a_{t-1}, \phi(S_{t-1}) = x_{t-1})}$$
$$= \frac{\mathbb{P}(S_t = s_t) \mathbb{P}(A_t = a_t | S_t = s_t)}{\mathbb{P}(\phi(S_t) = x_t) \mathbb{P}(A_t = a_t | \phi(S_t) = x_t)}, \tag{D.24}$$

or equivalently,

$$\frac{\mathbb{P}(A_t = a_t, S_t = s_t | A_{t-1} = a_{t-1}, \phi(S_{t-1}) = x_{t-1})}{\mathbb{P}(A_t = a_t, \phi(S_t) = x_t | A_{t-1} = a_{t-1}, \phi(S_{t-1}) = x_{t-1})}$$
$$= \frac{\mathbb{P}(A_t = a_t, S_t = s_t)}{\mathbb{P}(A_t = a_t, \phi(S_t) = x_t)}. \tag{D.25}$$

Since the original process $(S_t, A_t, R_t)_{t \ge 1}$ is an MDP, we have

$$\frac{\mathbb{P}(A_t = a_t, S_t = s_t | A_{t-1} = a_{t-1}, \phi(S_{t-1}) = x_{t-1})}{\mathbb{P}(A_t = a_t, \phi(S_t) = x_t | A_{t-1} = a_{t-1}, \phi(S_{t-1}) = x_{t-1})}$$
$$\times \mathbb{P}(S_{t+1} = s_{t+1} | A_t = a_t, S_t = s_t, A_{t-1} = a_{t-1}, \phi(S_{t-1}) = x_{t-1})$$
$$= \frac{\mathbb{P}(S_{t+1} = s_{t+1} | A_t = a_t, S_t = s_t) \mathbb{P}(A_t = a_t, S_t = s_t)}{\mathbb{P}(A_t = a_t, \phi(S_t) = x_t)},$$

leading to

$$\mathbb{P}(S_{t+1} = s_{t+1} | A_t = a_t, \phi(S_t) = x_t, A_{t-1} = a_{t-1}, \phi(S_{t-1}) = x_{t-1})$$
$$= \mathbb{P}(S_{t+1} = s_{t+1} | A_t = a_t, \phi(S_t) = x_t). \tag{D.26}$$

Equation D.26 implies that when $k = 1$, equation D.22 holds.

Furthermore, by summing over $s_{t+1} \in \phi^{-1}(x_{t+1})$ on both sides of equation D.26, we obtain that

$$\mathbb{P}(\phi(S_{t+1}) = x_{t+1} | A_t = a_t, \phi(S_t) = x_t, A_{t-1} = a_{t-1}, \phi(S_{t-1}) = x_{t-1})$$
$$= \mathbb{P}(\phi(S_{t+1}) = x_{t+1} | A_t = a_t, \phi(S_t) = x_t).$$

This together with equation D.26 yields

$$\frac{\mathbb{P}(S_{t+1} = s_{t+1} | A_t = a_t, \phi(S_t) = x_t, A_{t-1} = a_{t-1}, \phi(S_{t-1}) = x_{t-1})}{\mathbb{P}(\phi(S_{t+1}) = x_{t+1} | A_t = a_t, \phi(S_t) = x_t, A_{t-1} = a_{t-1}, \phi(S_{t-1}) = x_{t-1})}$$
$$= \frac{\mathbb{P}(S_{t+1} = s_{t+1} | A_t = a_t, \phi(S_t) = x_t)}{\mathbb{P}(\phi(S_{t+1}) = x_{t+1} | A_t = a_t, \phi(S_t) = x_t)} = \frac{\mathbb{P}(S_{t+1} = s_{t+1})}{\mathbb{P}(\phi(S_{t+1}) = x_{t+1})},$$

where the last equation again, follows from the backward-transition-irrelevance.

Under the stationarity assumption, it leads to

$$\frac{\mathbb{P}(S_t = s_t | A_{t-1} = a_{t-1}, \phi(S_{t-1}) = x_{t-1}, A_{t-2} = a_{t-2}, \phi(S_{t-2}) = x_{t-2})}{\mathbb{P}(\phi_2(S_t) = x_t | A_{t-1} = a_{t-1}, \phi(S_{t-1}) = x_{t-1}, A_{t-2} = a_{t-2}, \phi(S_{t-2}) = x_{t-2})}$$
$$= \frac{\mathbb{P}(S_t = s_t)}{\mathbb{P}(\phi(S_t) = x_t)}.$$

Applying the same arguments can be repeatedly for $t - 2$ times, we obtain that

$$\frac{\mathbb{P}(S_t = s_t | \{A_{t-k} = a_{t-k}, \phi(S_{t-k}) = x_{t-k}\}_{1 \leq k \leq t-1})}{\mathbb{P}(\phi_2(S_t) = x_t | \{A_{t-k} = a_{t-k}, \phi(S_{t-k}) = x_{t-k}\}_{1 \leq k \leq t-1})}$$
$$= \frac{\mathbb{P}(S_t = s_t)}{\mathbb{P}(\phi(S_t) = x_t)}. \tag{D.27}$$

Now, using the same arguments to equation D.24 and equation D.26, we obtain

$$\mathbb{P}(S_{t+1} = s_{t+1} | \{A_{t-k} = a_{t-k}, \phi(S_{t-k}) = x_{t-k}\}_{0 \leq k \leq t-1})$$
$$= \mathbb{P}(S_{t+1} = s_{t+1} | A_t = a_t, \phi(S_t) = x_t). \tag{D.28}$$

This proves equation D.22. It is immediate to see that equation D.22 yields

$$(A_{t-k}, \phi(S_{t-k}))_{1 \leq k \leq t-1} \perp\!\!\!\perp \phi(S_{t+1}) | (\phi(S_t), A_t),$$

which implies that the evolution of $\{\phi(S_t)\}_t$ is Markovian.

Next, we demonstrate that the reward function when confined to the abstract state space also satisfies the Markov property. Similar to equation D.25, by combining equation D.27 and equation D.16, we obtain that

$$\frac{\mathbb{P}(A_t = a_t, S_t = s_t | \{A_{t-k} = a_{t-k}, \phi(S_{t-k}) = x_{t-k}\}_{1 \leq k \leq t-1})}{\mathbb{P}(A_t = a_t, \phi(S_t) = x_t | \{A_{t-k} = a_{t-k}, \phi(S_{t-k}) = x_{t-k}\}_{1 \leq k \leq t-1})}$$
$$= \frac{\mathbb{P}(A_t = a_t, S_t = s_t)}{\mathbb{P}(A_t = a_t, \phi(S_t) = x_t)}. \tag{D.29}$$

Notice that in the original MDP, the reward function satisfies the Markov property, i.e., the conditional mean of the reward is independent of $\{A_{t-k}, \phi(S_{t-k})\}_{1 \leq k \leq t-1})$, given $A_t$ and $S_t$. Consequently, we can multiply the $\sum_r r \mathbb{P}(R_t = r | A_t = a_t, S_t = s_t)$ on both sides of equation D.29 and obtain that

$$\frac{\mathbb{E}[R_t \mathbb{I}(A_t = a_t, S_t = s_t)] | \{A_{t-k} = a_{t-k}, \phi(S_{t-k}) = x_{t-k}]\}_{1 \leq k \leq t-1}}{\mathbb{P}(A_t = a_t, \phi(S_t) = x_t | \{A_{t-k} = a_{t-k}, \phi(S_{t-k}) = x_{t-k}\}_{1 \leq k \leq t-1})}$$
$$= \frac{\mathbb{E}[R_t \mathbb{I}(A_t = a_t, S_t = s_t)]}{\mathbb{P}(A_t = a_t, \phi(S_t) = x_t)}.$$

By summing $s_t$ over $\phi^{-1}(x_t)$ on both sides of the equation, we obtain

$$\mathbb{E}(R_t | A_t = a_t, \phi(S_t) = x_t, \{A_{t-k} = a_{t-k}, \phi(S_{t-k}) = x_{t-k}\}_{0 \leq k \leq t-1})$$
$$= \mathbb{E}(R_t | A_t = a_t, \phi(S_t) = x_t).$$

This proves the Markov property of the reward function when restricted to the abstract state space. The proof is hence completed.

### D.5 LEMMA D.3 AND ITS PROOF

We first state Lemma D.3.

**Lemma D.3** *Suppose the reward is a deterministic function of the state-action pair. Then, the followings hold for both the bandit and MDP examples:*

- *The forward abstraction selects the first two groups $S_t^{(1)}$ and $S_t^{(2)}$;*

- *The proposed backward abstraction selects the last two groups $S_t^{(2)}$ and $S_t^{(3)}$;*
- *The proposed DSA selects their intersection $S_t^{(2)}$ and converges in two steps, resulting in a smaller subset of variables compared to the two non-iterative procedures.*

We next prove this lemma. Notice that reward-irrelevance requires the reward function (i.e., the conditional mean of the immediate reward given the state-action pair) to depend on the state only through its abstraction. Under the deterministic reward assumption in Lemma D.3, such a conditional mean independence is equivalent to conditional independence. In other words, reward-irrelevance is achieved if the reward is conditionally independent of the state given the action and the abstract state.

### D.5.1 PROOF FOR THE BANDIT EXAMPLE

We first consider the bandit example. As commented in the main text, in the contextual bandit setting, model-irrelevance is reduced to reward-irrelevance whereas backward-model-irrelevance is reduced to behavior-policy-irrelevance. Consequently, it is immediate to see that the assertions in the first two bullet points hold.

To prove the last bullet point, notice that according to the first bullet point, DSA would select $S_t^{(1)}$ and $S_t^{(2)}$ in the first iteration. In the second iteration, DSA would select $S_t^{(2)}$, due to the conditional independence between $A_t$ and $S_t^{(1)}$ given $S_t^{(2)}$. To verify such conditional independence, notice that there are two paths from $S_t^{(1)} \to A_t$: (i) $S_t^{(1)} \to R_t \leftarrow A_t$; (ii) $S_t^{(1)} \to R_t \leftarrow S_t^{(2)} \to A_2$. The second path is blocked by $S_t^{(2)}$ whereas the first path contains a collider $R_t$ which is a child of $S_t^{(2)}$. Consequently, both paths fail to $d$-connect $S_t^{(1)}$ and $A_t$ given $S_t^{(2)}$, leading to the desired conditional independence property. Since $R_t$ is a child of $S_t^{(2)}$, in the third iteration, $S_t^{(2)}$ will be selected as well. Similarly, in the subsequent iteration, $S_t^{(2)}$ will again be selected since $A_t$ is a child of $S_t^{(2)}$. Consequently, DSA converges after two iterations.

### D.6 PROOF FOR THE MDP EXAMPLE

As discussed in the main text:

- Selecting the first group of variables achieves reward-irrelevance.
- Selecting the last group of variables achieves behavior-policy-irrelevance.
- Selecting the second group of variables achieves both transition-irrelevance and backward-transition-irrelevance.

It is immediate to see that the the assertions in the first two bullet points hold. To prove the last bullet point, again, notice that DSA would select $S_t^{(1)}$ and $S_t^{(2)}$ in the first iteration. In the second iteration, DSA would select $S_t^{(2)}$, due to (i) the conditional independence between $S_t^{(1)}$ and $A_t$ given $S_t^{(2)}$ and (ii) that between $S_{t+1}^{(1)}$ and $(A_t, S_t^{(2)})$ given $S_{t+2}^{(1)}$. This is because (i) implies behavior-policy-irrelevance and (ii) implies backward-transition-irrelevance (see the discussion below equation 5) when restricted to the space of the first two groups of variables.

It remains to verify (i) and (ii). To prove (i), notice that all paths from $S_t^{(1)}$ to $A_t$ is either blocked by $S_t^{(2)}$, or include the collider $S_t^{(1)} \to R_t \leftarrow A_t$. To prove (ii), similarly, notice that all paths from $(A_t, S_t^{(2)})$ to $S_{t+1}^{(1)}$ is either blocked by $S_t^{(2)}$, or include the collider $S_{t+1}^{(1)} \to R_{t+1} \leftarrow A_{t+1}$.

Thus, we have shown that DSA would select $S_t^{(2)}$ in the second iteration. In the third iteration, notice that there is a path $S_t^{(2)} \to S_t^{(1)} \to R_t$ which is not blocked by $A_t$. Consequently, DSA would select $S_t^{(2)}$ in the third iteration as well. Similarly, in the subsequently iteration, DSA would select $S_t^{(3)}$, due to the path $S_t^{(2)} \to S_t^{(3)} \to A_t$. As such, it converges after two iterations.

