# OpenReview forum: "Off-policy Evaluation with Deeply-abstracted States"
_ICLR.cc/2025/Conference — ICLR 2025 Conference Withdrawn Submission_

### Official Review · Reviewer_CeyP · 2024-10-20

**Soundness:** 3
**Presentation:** 3
**Contribution:** 3
**Rating:** 5
**Confidence:** 2

**Summary:**

The paper "Off-Policy Evaluation with Deeply Abstracted States" addresses the challenge of off-policy evaluation (OPE) in large state spaces. Off-policy evaluation assesses the impact of a target policy using historical data collected under a different behavior policy. Large state spaces make this task difficult due to increased sample complexity. The authors propose state abstraction techniques for OPE, introducing a method to construct deeply abstracted states by iteratively projecting the original state space into smaller spaces. Their contributions include (1) defining irrelevance conditions for OPE, (2) creating an iterative procedure to compress the state space, and (3) proving the Fisher consistency of several OPE estimators when applied to abstract state spaces. The proposed abstractions are validated through numerical experiments, demonstrating improvements in sample efficiency and OPE accuracy.

**Strengths:**

The paper applies state abstraction, which has been widely studied for policy learning. This approach has the potential to significantly reduce state space cardinality and improve the accuracy of OPE estimators. The proposed iterative procedure for generating deeply abstracted states is a creative solution. It simplifies the sample complexity of OPE by shrinking the state space without compromising the representational capacity for policy evaluation. The method supports multiple types of OPE estimators, including value-based methods, importance sampling, and doubly robust methods. This shows the paper’s wide relevance across different off-policy evaluation scenarios. The authors provide rigorous theoretical backing, including irrelevance conditions and Fisher consistency proofs for various estimators, ensuring that the abstracted states do not distort the OPE process.

**Weaknesses:**

The iterative nature of the abstraction process, while powerful, might introduce computational overhead that is not fully addressed. Although the paper validates the approach with numerical experiments, the experimental section seems underdeveloped. There is a lack of detailed comparisons with baseline methods, especially in real-world scenarios. It would be helpful to see more empirical results across diverse datasets to better understand the practical performance of deeply abstracted states.

This paper presents a compelling theoretical advancement in state abstraction for OPE, but further exploration is needed to evaluate its practical impact and computational efficiency across varied domains.

**Questions:**

What is the computational overhead of the iterative state abstraction process, and how does it scale with increasingly large state spaces? Are there scenarios where the cost of abstraction outweighs the benefits in terms of sample complexity reduction?

Can the proposed methodology be easily integrated into existing OPE frameworks or reinforcement learning pipelines? What modifications would be needed for real-world application?

---

> ### Author Response · Authors · 2024-11-20
>
> We greatly appreciate your valuable assessment of our work, many of which will lead to a more readable and self-contained version of our paper. Below, we address certain specific concerns.
>
> --**Computational overhead**. Thank you for pointing this out. We completely agree that the iterative procedure will incur some computational cost. However, we have found that performing the iteration just four times is sufficient to achieve a noticeable increase in accuracy. As a result, the computational burden is relatively small compared to the improvement in accuracy. Additionally, we believe there are methods to optimize the computation in the future.
>
> --**Integration into existing OPE frameworks**. Thank you for your comment. Indeed, the proposed method can be easily integrated into existing OPE frameworks, as detailed in Appendix B.

---

### Official Review · Reviewer_kAt8 · 2024-10-26

**Soundness:** 3
**Presentation:** 1
**Contribution:** 1
**Rating:** 3
**Confidence:** 4

**Summary:**

The paper proposes a set of conditions on state abstractions in order to effectively use the state abstraction for off-policy evaluation. It then proposes an iterative procedure to learn abstractions that satisfy these conditions, and shows that the learned abstractions result in accurate OPE for various OPE estimators.

**Strengths:**

- The paper nicely unifies several ideas such as referencing prior work on model-irrelevant and pi-irrelevant abstractions. And also relating its backward model to prior work that learns an inverse dynamics model.
- The results are for a wide class of OPE estimators compared to prior work that discusses OPE and abstractions only for a particular OPE estimator.
- The work tackles a problem that has received relatively little attention.

**Weaknesses:**

- The motivation of the paper is unclear. There is a possibly interesting idea here in the backward model, but it is unclear what the purpose of this model is. For example, Lemma 1 suggests that the previous irrelevance conditions are sufficient for consistent OPE. So it is unclear to me why we need an alternative condition to also get consistency. From theorem 1, I see that the backward model also gives us two additional irrelevance conditions, but why do we need these if all we care about is consistent OPE? Is the backward model condition easier to learn? The empirical method seems to combine both forward and backward models, but is there a reason to do so? Why is only one of them insufficient? This seems core to the paper and needs to be made clearer.
- The paper may need a big rewrite.
  - The main paper has an extremely sparse empirical section, which is a weakness given that the paper is pitched as making an empirical contribution.
  - The paper is also missing a discussion and conclusion section.
  - The introduction immediately uses terms such as backward model or time-reversed MDP as if their meanings are obvious but they are not necessarily so and need to be clear.
  - It is unconventional to write the Q-function as Q(a, s) vs. Q(s,a) (Line 155). There are several instances of switching the notation.
$\psi$ is not defined although its meaning can be inferred based on Figure 1.
  - Line 240 has a missing section reference.
  - The meaning of “identifiable” (line 247) needs to be specified.
- The empirical details need to be improved. Section 5 does not include any details of the loss functions (they are in the appendix) which is critical given that the paper is pitched as also making an empirical contribution. Section 5 also includes results only on one domain, which is highly insufficient to make a general claim about the method. There are also no details on the number of trials, seeds, confidence intervals etc.

**Questions:**

See question above on why backward model is needed and why Definition 6 and 7 are needed given other assumptions are sufficient for consistent OPE.

---

> ### Author Response · Authors · 2024-11-18
>
> We greatly appreciate your valuable assessment of our work. Below, we address certain specific concerns.
>
> --**Backward model irrelevance**. Since there are different OPE methods, backward model irrelevance is employed to ensure the ($\rho^{\pi}$) $w^{\pi}$ irrelevance while the forward model irrelevance is used to ensure the $Q^{\pi}$ irrelevance, which is the key for the (SIS) MIS method and Q-function-based method, respectively.
>
> What we propose is **an iterative method**, as highlighted in the title. Our approach alternates between forward and backward abstraction on the state space obtained from the previous iteration. Each iteration ensures that the cardinality of the state space does not increase, effectively maintaining or reducing complexity. Consequently, this iterative procedure progressively reduces the state cardinality, ultimately yielding a **deeply abstracted state**. We refer to this approach as **deep state abstraction (DSA)**.
>
> Simulation results demonstrate that this iterative method achieves the smallest absolute bias and mean squared error (MSE), validating the effectiveness of our approach.
>
> --**Typos**. Thank you for pointing this out. We will correct all the typos in the revised paper and carefully proofread the paper in future versions to ensure accuracy.

---

> > ### Comment · Reviewer_kAt8 · 2024-11-20
> >
> > Thanks for your response. OK this somewhat clears things up. I will increase the score slightly, but I am still leaning on a reject due to the following reasons: 1) the empirical section needs to be much more thorough, which I dont think can be adequately done in this review cycle (more analysis, more environments etc); again the primary reason being that this is not pitched as a theory paper, so it should have a strong empirical component. 2) Incomplete discussion/conclusion sections. Overall, I think the work is heading in an interesting direction and may be more suitable for publication in its next iteration.
> >
> > I'd also encourage the authors to see: Abstract Reward Processes: Leveraging State Abstraction for Consistent Off-Policy Evaluation. Chaudhari et al. 2024, which may also be helpful in extending the presented ideas more.

---

### Official Review · Reviewer_HUmY · 2024-10-31

**Soundness:** 3
**Presentation:** 2
**Contribution:** 2
**Rating:** 5
**Confidence:** 3

**Summary:**

The authors propose a method to perform state abstraction for the purpose of off-police evaluation using deep learning. Their method optimizes two notions of state similarity iteratively. The authors also prove theoretical consistency of their results.

**Strengths:**

1. The authors propose the novel idea of backwards model irrelevance.

2. The paper is well organized, and provides a lot of background and explanations.

**Weaknesses:**

1. Despite the authors' attempts at making things clearer, I didn't fully understand the notion of backwards model irrelevance, and why it would generally require more than one iteration with the forward model irrelevance. I think the paper would benefit from making this point clearer somehow. I found the examples on page 9 more confusing than helpful.

2. The example in the experiments looks very artificial. While it does show improved performance for the author's method, it being tailored to the the problem raises the question - how important is this problem in general. OPE is widely researched, but state abstraction for OPE is usually harder to justify, I'm not convinced by the experiment that it's a useful or significant technique. I would suggest giving one non-state-abstraction baseline, even if it outperforms the current results. Also, I think it would be good to run on another environments given in other OPE papers.

**Questions:**

1. Can you say anything about the rate of convergence for the iterative process?

2. What's the true meaning of backwards model irrelevance and why does it make sense to iteratively apply it with forward model irrelevance?

3. How are your results compared to non-state-abstraction methods?

4. This is probably in the paper but I couldn't find it - how many iterations are required and when do you stop iterating? It would have made sense for me if you included this in the pseudu-algorithm in the beginning of Subsection 4.3.

---

> ### Author Response · Authors · 2024-11-20
>
> We greatly appreciate your valuable assessment of our work, many of which will lead to a more readable
> and self-contained version of our paper. Below, we address certain specific concerns.
>
> --**Comparison to non-state-abstraction methods**. In fact, we have compared to one of the non-state-abstraction methods denoted as FQE in our simulation parts. Furthermore, we have claimed clearly that " Both figures show that the baseline FQE applied to the
> ground state space performs the worst among all cases. This comparison reveals the usefulness of state abstractions for OPE".
>
> --**Backwards model irrelevance**. The concept of backward model irrelevance refers to our observation that the state is tied to the behavior policy.
>
> In the context of our first toy example, confounder selection serves as a special case of our problem under certain conditions:(i) The state transition is independent, effectively transforming the MDP into a contextual bandit; (ii) The action space is binary, with the target policy consistently assigning either action 0 or action 1, aimed at assessing the average treatment effect; (iii) State abstractions are confined to variable selections. It is worth noting that this formulation mirrors the iterative confounder selection procedure discussed in Guo et al. (2022).  While our proposed iterative procedure shares similar spirits with the aforementioned
> algorithms, it addresses a more complex problem involving state transitions. Additionally, our focus
> is on abstraction that facilitates the engineering of new feature vectors, rather than merely selecting a
> subset of existing ones. Consequently, such an iterative
> procedure progressively reducing state cardinality, which ultimately yields a deeply-abstracted state. We thus refer to our approach as deep state abstraction (DSA). The simulation results indicate that such iterative method can get the smallest absolute bias and MSE.
>
>
>
> --**Reference**.
> - F Richard Guo, Anton Rask Lundborg, and Qingyuan Zhao. Confounder selection: Objectives and
> approaches. arXiv preprint arXiv:2208.13871, 2022.

---

> > ### Comment · Reviewer_HUmY · 2024-11-21
> >
> > Thank you for the reply.

---

### Official Review · Reviewer_xehG · 2024-11-03

**Soundness:** 2
**Presentation:** 2
**Contribution:** 3
**Rating:** 3
**Confidence:** 4

**Summary:**

The paper studies the use of state abstractions, learned through forward and backward losses, for the task of off-policy evaluation (OPE). It outlines the conditions that the abstractions must satisfy to enable the application of OPE methods to a more compact abstract MDP. Specifically, the model-irrelevance condition is adapted for policy evaluation and applied to both forward and backward views of the MDP. By sequentially applying learning rules for forward and backward model irrelevance conditions, a two-step abstraction learning procedure is developed. This procedure results in a compact MDP, making it possible to apply standard OPE methods effectively.

**Strengths:**

- The irrelevance conditions proposed in [1] are adapted well to impose irrelevance of components of OPE methods (like the IS and MIS ratios).
- The final result, that OPE methods can be applied directly to the abstract MDP obtained after the two-step abstraction learning process, is a useful result. Effective two-step abstraction learning methods can make OPE practically applicable for problems with high dimensional states.

---
[1] Li, Lihong, Thomas J. Walsh, and Michael L. Littman. "Towards a unified theory of state abstraction for MDPs." AI&M 1.2 (2006): 3.

**Weaknesses:**

- Fisher consistency: The statement of the results claim to show Fisher consistency of the corresponding abstract-state estimators, however, the proofs are for unbiasedness and identifiability. It is unclear whether these two conditions imply Fisher consistency.
- Experiments:
  - [L511] states that the reported metrics are MSE and absolute bias, however, Figure 5 reports relative MSE and relative absolute bias. The latter has not been defined.
  - The abstraction learning itself requires some amount of data that aids the OPE performance, which must also be reported.
     - The number of data points for OPE is very less (10-60 trajectories) and the plots must be extended further to the right for a meaningful comparison for the baseline methods to also receive sufficient data.
  - The error bars are not defined (standard error? confidence intervals?) and are computed over only 20 trials.

Nitpicks:
- All the irrelevance conditions are a simple adaptation of the ones in [1] and the paper’s claim on *defining* these conditions for OPE is overclaiming.

**Questions:**

- Can the authors more rigorously specify why their proof sketches imply Fisher consistency of their methods? Moreover, are the OPE methods (without abstraction) themselves Fisher consistent?
- How much data is required for learning abstractions in the experiments with the two-step procedure?
- A comparison of MSE with importance-sampling (and not just FQE) would be insightful, particularly for larger dataset sizes, would be insightful. Being an unbiased OPE method, it would be interesting to see up till what point the variance reduction afforded by abstractions is beneficial despite the bias incurred.

---

> ### Author Response · Authors · 2024-11-18
>
> We greatly appreciate your valuable assessment of our work. Below, we address certain specific concerns.
>
> --**Fisher consistency**.  *Fisher consistency is described as “Roughly this requires that if the whole ‘population’ of random variables is observed, then the method of estimation should give exactly the right answer"*, see Cox and Hinkley (1974, p. 287) , Tasche (2017). Thus,
> $\mathbb{E} [f_1(Q^{\pi})]=\mathbb{E} [f_2(\rho^{\pi})]=\mathbb{E} [f_3(w^{\pi})]=\mathbb{E}[f_4(Q^{\pi},w^{\pi})]=J(\pi)$
> shows that the OPE methods (without abstraction) themselves are Fisher consistent. For our methods,
> - $J(\pi)=\mathbb{E} [f_1(Q^{\pi})]=\mathbb{E} [f_1(Q^{\pi}_{\phi})]$ implies the Q-function-based method is Fisher consistent;
> - $J(\pi)=\mathbb{E} [f_3(w^{\pi})]=\mathbb{E} [f_3(w^{\pi}_{\phi})]$ means MIS is Fisher consistent;
> - $J(\pi)=\mathbb{E} [f_2(\rho^{\pi})]=\mathbb{E} [f_2(\rho^{\pi}_{\phi})]$ indicates SIS is Fisher consistent;
> - $J(\pi)=\mathbb{E}[f_4(Q^{\pi},w^{\pi})]=\mathbb{E} [f_4(Q^{\pi}_{\phi},w^{\pi} _{\phi})]$  describes Fisher consistency of the double robust method.
>
> This logic can also be found in Park and Weisberg (1998), equation (6) in page 232.
>
> --**Two-step procedure**. We find this comment regarding the 'two-step procedure' confusing. The two-step approach refers to our earlier version submitted to NeurIPS. For this conference, we submitted a modified version, yet the term appears twice in the reviewer’s comments. A similar concern was raised by one of the reviewers in our previous submission. This reviewer, who provided the same score, had expressed a willingness to revise their score based on updates to our proof.
>
> In response, we devoted significant effort during the rebuttal period to provide a clear and accessible proof within the word limit. Unfortunately, during the discussion period, the reviewer became reluctant to engage, only responding in the final hours. From their last response, we understood that their primary concerns were addressed. However, they did not specify which aspects of our responses were inadequate but maintained their original score, stating that "a substantial revision is needed."
>
> Additionally, we would like to clarify that we did not perform data splitting to train abstractions and OPE estimators. Both procedures were conducted on the same dataset, ensuring no reduction in sample size that could compromise the performance of the final estimator.
>
> --**Reference**:
> - Cox, D. R., & Hinkley, D. V. (1974). Theoretical statistics. Chapman and Hall, London.
> - Park, C., &  Weisberg, S. (1998). Fisher consistency of GEE models under link misspecification. Computational statistics & data analysis, 27(2), 229-235.
> - Tasche, D. (2017). Fisher consistency for prior probability shift. Journal of Machine Learning Research, 18(95), 1-32.

---

> > ### Comment · Reviewer_xehG · 2024-11-20
> >
> > Thanks for the response. A couple of pertinent questions still stand:
> >
> > > Fisher consistency
> >
> > As per the definition provided by the authors above, Fisher consistency is a "population limit" guarantee. The estimator being equal in expectation (with finite data) is the property of *unbiasedness*. Can the authors elaborate more on: 1) whether they agree these two are different properties of estimators, and 2) which of these two properties their proofs imply? My read of the paper suggests that the final statement (expected value equaling the expected return) implies unbiasedness.
> >
> > >  Data splitting
> >
> > Thanks for the clarification. The question, put differently, is: Compared to FQE, additional data is required by this procedure to *learn* the abstractions. How much data is that? Is is really just 10 trajectories that are used to both learn abstractions and perform OPE?
> >
> > Additionally, the comment about additional comparison with IS-like methods (expected to incur lower bias) is unaddressed. I will maintain my score.

---

### Note · Authors · 2024-11-25

I have read and agree with the venue's withdrawal policy on behalf of myself and my co-authors.